# Efficient lateral-structure perovskite single crystal solar cells with high operational stability

Yilong Song[1], Weihui Bi[1], Anran Wang[1], Xiaoting Liu[1], Yifei Kang[1] & Qingfeng Dong[1]*

The power conversion efficiency of perovskite polycrystalline thin film solar cells has rapidly increased in recent years, while the stability still lags behind due to its low thermal stability as well as the fast ion migration along the massive grain boundaries. Here, stable and efficient lateral-structure perovskite solar cells (PSCs) are achieved based on perovskite single crystals. By optimizing anode contact with a simple surface treatment, the open circuit voltage and fill factor dramatically increase and promote the efficiency of the devices exceeding 11% (0.05 to 1 Sun) compared to that of 5.9% (0.25 Sun) of the best lateral-structure single crystal PSCs previously reported. Devices show excellent operational stability and no degradation observed after 200 h continuous operation at maximum power point under 1 Sun illumination. Devices with scalable architectures are investigated by utilizing interdigital electrodes, which show huge potential to realize low cost and highly efficient perovskite photovoltaic devices.

---

[1] State Key Laboratory of Supramolecular Structure and Materials, College of Chemistry, Jilin University, Changchun, China. *email: qfdong@jlu.edu.cn

Organic–inorganic hybrid perovskite materials attracted great attention for the rocketing developed power conversion efficiency (PCE) in photovoltaics, which had risen rapidly from the initial 3.81%[1] to 25.2%[2] in the last 10 years, but the stability issues still limited the commercialization of perovskite solar cells (PSCs). Currently, the most efficient PSCs were based on perovskite polycrystalline (PC) thin films[3–10]. However, the massive amorphous or low crystallinity grain boundaries in PC thin films were proved as the main reasons for the low thermal decomposition temperature[11] and high speed pathways for ion migration[12–14], which were two major problems for the stability issue of PSCs. Compared with PCs, perovskite single crystals (SCs) were immunized from grain boundaries, have shown dramatically enhanced optoelectronic properties, such as longer carrier diffusion length, lower trap densities[15], extended absorption spectrum[15,16] and suppressed ion migration effect[12], which guarantee superior device performance and provide an effective way to improve the stability of the PSCs without sacrificing efficiency. Currently, a significantly enhanced thermal decomposition temperature up to 240 ºC was reported in SC which was much higher than that of 150 ºC in perovskite PC thin films[11]. In addition, better chemical stability in atmosphere and longer storage time in air have also been reported in perovskite SCs[11,15,17,18]. As a result, perovskite SCs showed great potential for achieving SC-PSCs with both high efficiency and long-term stability. Since the first efficient SC-PSC based on bulk methylammonium lead iodide (MAPbI$_3$) SC was reported in 2016[19], the efficiency of SC-PSCs was rapidly approaching to that of PC-PSCs[20,21]. However, the long-term operational stability of SC-PSCs have not been investigated yet[6,16,20,22–25], because SC-PSCs also brought additional stability issues in sandwiched structure, which was widely used in current PC-PSCs. For example, the large discrepancy of thermal expansion coefficients and mechanical properties between SC and glass substrate would lead to the peeling off the SC from the substrate during temperature variation at operational conditions.

Great efforts were made towards the emerging efficient lateral structure PSCs because it was believed as a prerequisite structure for efficient interdigitated back contact (IBC) structure PSCs to realize high performance and low-cost devices[26–32]. Stand-free lateral structure SC-PSC immured from strain stress induced mechanical device degradation due to different thermal expansion coefficients for multiple functional layers as well as substrates, which potentially increased the device stability than regular sandwich structures. The lateral structure was also a stand-free structure which did not require expensive indium tin oxide (ITO) substrate compared with traditional sandwiched-structure PSCs[33]. The lateral structured PSCs could further promote the light absorption efficiency because light could directly illuminate on crystal surface without losing caused by absorption of glass substrate as well as conductive electrode, which would contribute to the enhanced photocurrent and efficiency than regular PSC based on ITO or fluorine-doped tin oxide (FTO) electrode. Although the best SC-PSC was still based on sandwich structure with expensive ITO, an outstanding PCE of 21.09% and fill factor (FF) up to 84.3% was achieved in a 20 μm thick MAPbI$_3$ single crystals device, which showed huge potential for developing high performance PSCs based on perovskite single crystal[20]. The efficient lateral-structure SC-PSCs were realized based on MAPbI$_3$ SCs by piezoelectric poling with a PCE of 1.88%[19], but the poling-generated grain-boundaries sacrificed SC device performance. Recently, by optimizing cathode contact, the efficiency was significantly increased to 5.9% without destroying SC structure[34]. However, there was still challenge to realize selective contact at the anode of the device for higher efficiency

lateral-structure SC-PSCs, due to the large energy level mismatch between anode and perovskite[19,34,35].

In this work, highly stable SC-PSCs are achieved in lateral-structure devices with optimized anode contact by a simple methylammonium iodide (MAI) surface treatment, which is found to be an effective way to enhance performance due to the passivation effect and here we explore it for lateral solar cells[16]. An ultra-thin MAI layer is introduced to promote the surface potential of perovskite single crystal by ~80 meV towards the valance band, which also significantly enhances device efficiency. The resulting better energy level alignment at the anode contact result in dramatically enhanced open circuit voltage ($V_{OC}$) and FF in SC-PSC devices. Besides, the surface treatment also leads to a well passivated and highly conductive SC surface, which enables efficient charge collection in lateral structure devices with electrodes spacing up to 50 μm. A PCE exceeding 11% is achieved at different light intensity from 0.05 to 1 Sun in MAI-treated PSCs compare to that of 5.9% (0.25 Sun) of the best lateral-structure SC-PSCs previously reported[34]. Devices show excellent operational stability and no degradation is observed after 200 h continuous operation at maximum power point (MPP) under 1 Sun illumination. At the same time, scalable architectures for large-area SC-PSCs are realized by simply utilizing interdigital electrodes. The lateral-structure SC-PSCs, combining ITO-free low-cost device structure, high efficiency and inspiring device stability, show huge potential to realize low cost and highly efficient perovskite photovoltaic devices.

## Results

**Fabrication of lateral structure PSCs.** The lateral-structure SC-PSCs could be simply fabricated by scalable architecture designed as Fig. 1a. Anode electrode was deposited on the single crystal surface (Fig. 1b) by shadow mask for the first step. Then electron transport layer (ETL) of C60/BCP was thermally evaporated on crystal surface followed by cathode deposited on the ETL for the final device. Figure 1c–d were the photographs of the integrate device. The width of each finger of electrode was 1.5 mm in interdigital electrodes and 1 mm in single device and the spacing between each anode and cathode was about 65 μm in interdigital electrodes (Fig. 1a) and about 50 μm in a single device (Fig. 1e), which could be achieved simply by shadow masks and did not require expensive and complicated photolithography process.

**Surface treatment on perovskite single crystal.** In currently used lateral structured SC-PSCs, the anode was in direct contact with the surface of the SC, and the work function of Au anode differs greatly from that of perovskite. To achieve better interfacial energy level alignment, trace amount of MAI was introduced by simply spin-coating diluted MAI solution on the surface of SCs to change the proportion of MAI and PbI$_2$ content on MAPbI$_3$ surface. As measured by Kelvin probe force microscopy (KPFM) with multiple samples, the surface potential of the SC after treatment with MAI increased by an average value (Table 1) of about 80 meV compared to the untreated ones[36,37] which resulted in the better energy level matching between the single crystal surface and Au anode without affecting cathode contact because of the inserted C60/BCP interfacial layer (Fig. 2a–e). In order to get accurate KPFM results and suppress the influence caused by surface composition, dipole moment, ferroelectricity and ion motion, all the samples for KPFM test were got from the same surface of the same piece of single crystal. Different metals of Cu and Au were deposited on the same single crystal as reference to double confirm the variation of single crystal surface potential with multiple scans at different positions.

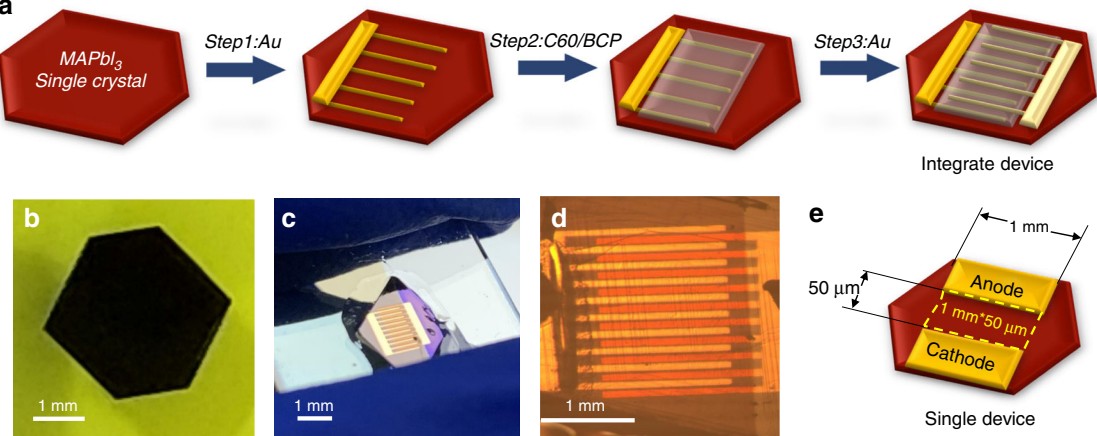

**Fig. 1 Fabrication of lateral structure perovskite solar cells. a** Schematic diagram of preparation process of large-area lateral structure perovskite single crystal solar cells. **b** Image of the MAPbI₃ single crystal. **c** and **d** Photographs of the lateral-structure SC-PSCs. **e** Structure of a regular single device with area of 50 μm × 1 mm. The large-area device was integrated by 19 single devices (65 μm × 1.5 mm) with total device area of 1.85 mm².

**Table 1 Statistical table of MAPbI₃ single crystal surface potential before and after MAI treatment by using Au and Cu as reference electrodes.**

|  | SC-Au (meV) | MAI/SC-Au (meV) | SC-Cu (meV) | MAI/SC-Cu (meV) |
|---|---|---|---|---|
|  | 250 | 150 | 420 | 290 |
|  | 200 | 125 | 430 | 325 |
|  | 150 | 75 | 400 | 275 |
|  | 200 | 100 | 450 | 300 |
|  | 200 | 125 | 415 | 315 |
| Average | 200 | 115 | 423 | 301 |
| Standard deviation | 33.35 | 28.50 | 18.57 | 19.81 |

In addition to the better band alignment, passivation effect was also observed after MAI treatment, which effectively recovered the surface damage of SCs caused by the fast re-dissolving for highly soluble methylammonium cations (MA⁺) by residual solution, as confirmed by both photoluminescence (PL) spectrum and time-resolved PL (TRPL) measurements. Figure 3a was the PL spectrum of the MAPbI₃ SCs before and after MAI treatment. The PL peak located at 789 nm for untreated SCs surface, and there was an 11 nm blue shift after MAI treatment, which indicated suppressed surface trap density[38,39]. Charge recombination lifetime of MAPbI₃ SCs were evaluated by TRPL measurement for SC before and after MAI treatment as shown in Fig. 3b. The MAI-treated SCs showed a significantly longer lifetime of 468 ns than 263 ns of that of the untreated SCs. The passivation of surface defects was one of the main factors for achieving efficient SC-PSCs, especially for lateral-structure devices with much longer carrier transit distance compared to the sandwich structured ones.

X-ray photoelectron spectroscopy (XPS)[40], Fourier transform infrared (FTIR)[41] spectroscopy, and time-of-flight secondary ion mass spectroscopy (TOF-SIMS)[42,43] measurements were taken for attaining further chemical composition information to show how MAI interact with MAPbI₃ surface as shown in Fig. 3c–g. The metallic lead was detected on surface of as grown MAPbI₃ single crystal as shown in XPS result, which was believed as a main factor of surface traps. After MAI treatment, the peak from metallic lead was effectively suppressed and a weak MAI peak emerged in XPS spectrum, which was also evidenced by the increased FTIR response around 3400 cm⁻¹ directly related to

N–H stretching vibrations for the MAI. TOF-SIMS measurement was also taken to reveal the surface and bulk properties of as grown and MAI-treated MAPbI₃ SCs in detail. The MAI-perovskite interaction was mainly a surface reaction instead of bulk evidenced by the TOF-SIMS study for both as grown and MAI-treated SCs. The lower crystalized SC surface, with higher initial second ion intensity, was found mainly on the top 5 nm near surface of SCs evidenced by MAI and PbI₂ related signals in TOF-SIMS results as shown in Fig. 3g. Based on the XPS, FTIR, and TOF-SIMS results, the very thin lower crystalized surface of SCs within 5 nm with metallic lead was detected and believed as a main origin of surface traps, MAI treatment contributed effectively passivated SC surface by suppressed formation of metallic lead.

It was worth noting that there was no detectable change of the surface morphology as well as crystallinity of SCs, as observed by scanning electron microscopy (SEM) and X-ray diffraction (XRD) measurement, indicating the MAI treatment did not destroy the single crystal structure. The SEM images of the surface without MAI treatment and with MAI treatment of the SCs were shown in Fig. 4a–b. There were flat surfaces and no apparent grain boundaries on surface of SCs before and after MAI treatment. XRD measurement was also used to study the crystalline quality of the surface in treated MAPbI₃ SCs (Fig. 4c), the high crystallinity of the crystal was demonstrated by the very sharp diffraction peak[44] without little changes before and after MAI treatment. Both SEM and XRD results indicated that the MAI treatment did not destroy the single crystal structure of perovskites.

The MAI treatment contributed significantly to the decreased surface trap density and increased carrier mobility across the surface of the MAPbI₃ SCs, which were evaluated by space-charge-limited current (SCLC) measurements[45,46] as shown in Fig. 4d. For SCLC method in lateral devices, a gap-type device was fabricated using thin single crystal with C60/BCP layers on top and Au electrodes with channel width of 50 μm[35,47]. From this J-V curve, we could obtain the carrier mobility for gap-type devices by the Geurst's SCLC model[35,47]:

$$I = \frac{2\mu\varepsilon_0\varepsilon_r W}{\pi}\left(\frac{V^2}{L^2}\right) \qquad (1)$$

According to the Geurst theory the threshold voltage for a

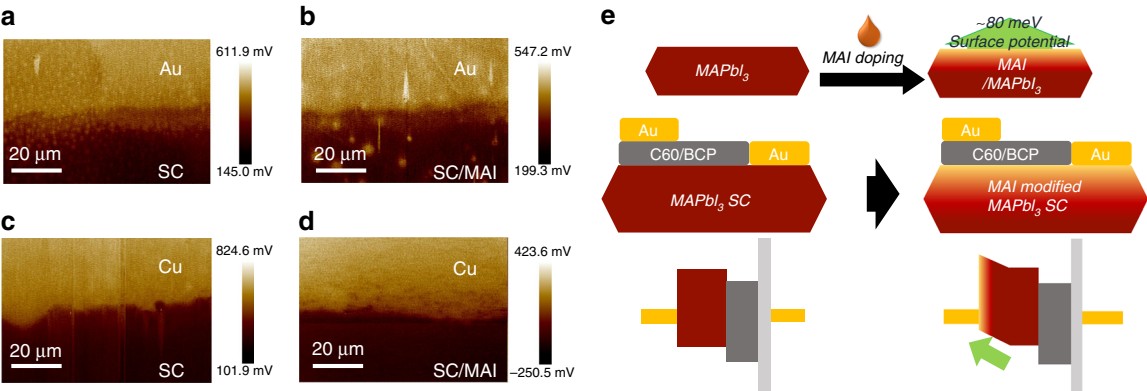

**Fig. 2 Surface treatment on perovskite single crystal. a–d** The KPFM images of the surface of single crystals before and after MAI treatment. **e** Schematic diagram of device structures and energy levels for SC-PSCs without and with MAI treatment.

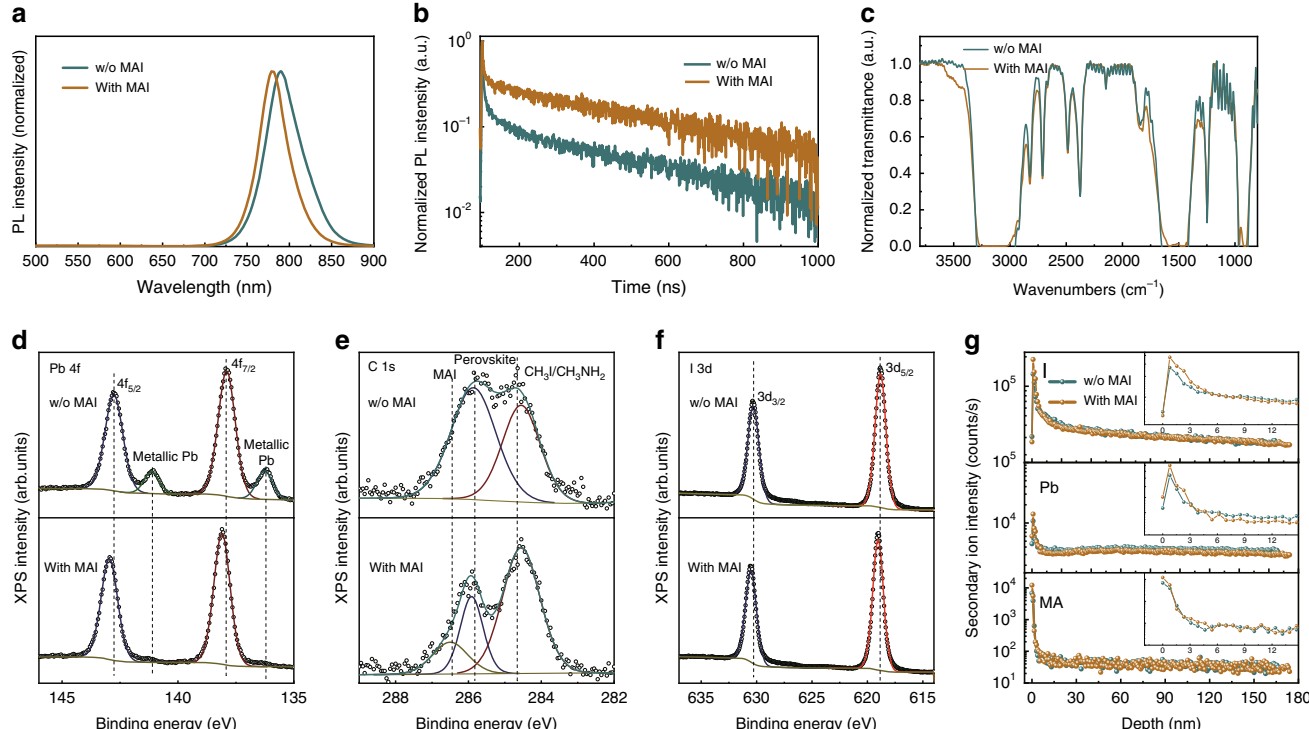

**Fig. 3 Passivation of single crystal surface by MAI treatment.** The PL (**a**), TRPL (**b**) FTIR (**c**), XPS (**d–f**), TOF-SIMS (**g**) spectra for MAPbI₃ single crystal surfaces without and with MAI treatment.

gap-type structure was equal to[47]

$$V_T = \frac{\pi \sigma_0 L}{4 \varepsilon_0 \varepsilon_r} \qquad (2)$$

where $\mu$ is the carrier mobility, $\sigma_0$ is the surface charge density per unit area, $\varepsilon_0$ is the vacuum permittivity and $\varepsilon_r$ is relative dielectric constant of perovskite, 32[15]. $W$ and $L$ are the device width and interelectrode distance, respectively.

After MAI treatment, the electron mobility of the SC was effectively enhanced to 1.16 cm² V⁻¹ s⁻¹ from that of 0.60 cm² V⁻¹ s⁻¹ in SC without treatment. The surface trap density was also significantly suppressed from $6.67 \times 10^9$ cm⁻² to $4.51 \times 10^9$ cm⁻² after MAI treatment.

The improvement of the optical-electrical properties of SC and devices with MAI surface treatment were also investigated by studying photocurrent and dark current of both SC-PSC structure devices (Fig. 5a) and devices with symmetric electrodes

(Fig. 5b–c). There was significantly enhanced rectifying behavior in MAI-treated SC-PSC device and bought a sharp increase in FF of PSCs from 32.5 to 55.1%, which will be further discussed below, indicated that the surface treatment resulted in an optimized electrode contact. The reverse dark current was also reduced for MAI-treated sample compared to the untreated one in a wide range of bias, which indicated the formation of more ideal diode structure. A significant enhancement of conductivity observed in lateral-structure devices with symmetrical gold electrodes both in dark and under light illumination proved the effective MAI treatment could reduce the voltage loss during current transport in lateral direction.

**Characterization of single crystal PSCs.** Intensity dependence of short circuit current ($J_{SC}$) and $V_{OC}$ curves were shown in Fig. 6a, b with light intensity in range of 0.0005–1.5 Sun. The ideal factor ($n$) was a vital indicator that reflected the charge recombination

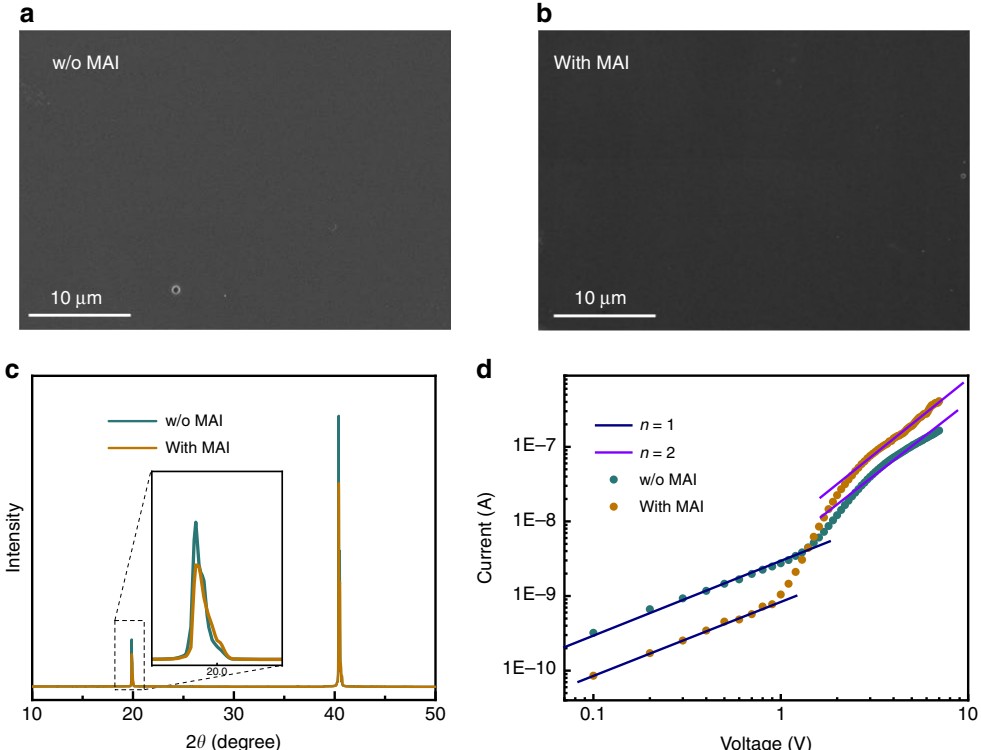

**Fig. 4 Characterization of single crystals surface.** SEM images of the MAPbI$_3$ single crystal surface **a** without and **b** with MAI treatment. **c** XRD patterns of MAPbI$_3$ single crystal, **d** SCLC without and with MAI treatment.

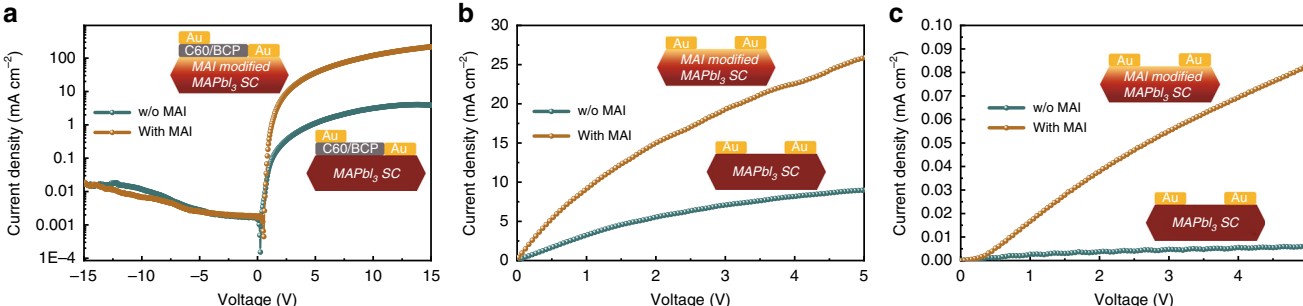

**Fig. 5 The photocurrent and dark current of lateral-structure devices. a** Dark current test of the lateral-structure device. **b** Photocurrent and **c** dark current measurements of lateral-structure devices at symmetrical gold electrodes.

in working devices[48]. The $n$ could be calculated by the intensity dependence of $V_{OC}$ curves, and the curves were fitted using the following equation[49]:

$$V_{OC} = \frac{n\mathrm{K}T}{q} \ln\left(\frac{J_{SC}}{J_0}\right) \qquad (3)$$

By calculation, $n = 2.3$ (with MAI) and 3.1 (control), respectively. For control device, the higher value of $n$ was gained that was attributed to the presence of severer nonradiative trap-assisted recombination[50]. And this proved that MAI had a marvelous passivation effect.

Benefiting from the better band alignment and passivation effect, the MAI surface treatment process led to considerable enhancement in device performance of SC-PSCs. Light intensity dependent $J$-$V$ curves of lateral structured SC-PSCs under AM1.5G illumination with different light intensity from 0.05 to 1.5 Sun by an AAA grade solar simulator, which was a standard test condition for regular PSCs, were shown in Fig. 6c–d. The PSC based on as grown SCs showed much faster degradation rate in photocurrent density under

higher light intensity (Fig. 6a). As a strong contrast, PSC with MAI treatment kept liner relationship between photocurrent density and light intensity and a PCE of 11.52% were achieved under 1 Sun condition without obvious hysteresis effect observed (Supplementary Fig. 1) and a champion PCE of 12.76% under 0.5 Sun benefited from the suppressed surface recombination. It was also worth to be noted that our crystal growth as well as device fabrication were processed in air condition. There was no much difference for $J$-$V$ curves of devices tested in vacuum or in air (Supplementary Fig. 2). A PCE of 4.03% with a $J_{SC}$ of 16.32 mA cm$^{-2}$, a FF of 32.5% and a $V_{OC}$ of 0.76 V were achieved in the control device without MAI treatment. After MAI treatment, there was a dramatical improvement in PCE, which was attributed to the significantly improved $V_{OC}$ and FF. The champion device with MAI treatment showed a PCE of 11.52% under 1 Sun with a $J_{SC}$ of 22.49 mA cm$^{-2}$, a FF of 55.1% and a $V_{OC}$ of 0.93 V. The increase of $V_{OC}$ was mainly due to the change of work function on the surface of single crystal and the optimized energy level alignment which greatly reduced the voltage loss. At the same time, the increased conductivity across SC surface

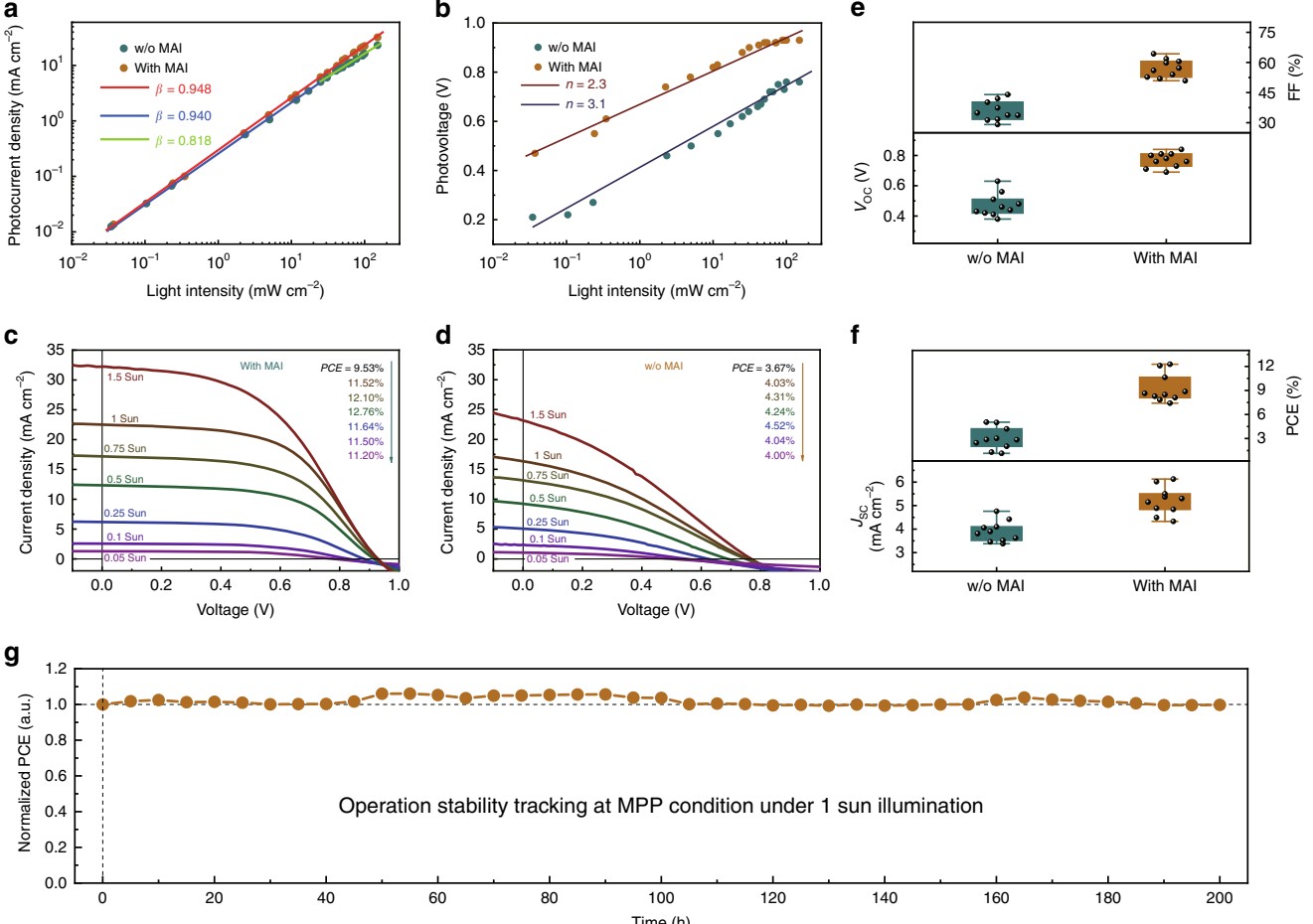

**Fig. 6 Device performance of lateral-structure solar cells. a** Light intensity (0.0005–1.5 Sun) dependence of $J_{SC}$ curves. **b** Light intensity (0.0005–1.5 Sun) dependence of $V_{OC}$ curves. $J$-$V$ curves of different light intensity with (**c**) and without (**d**) MAI treatment. **e** and **f** Statistical data of device performance before and after MAI treatment (0.25 Sun). **g** Long-term stability under continuous output at MPP condition (1 Sun).

further reduced $V_{OC}$ loss during the long range transport of charge carriers. The increased conductivity also led to the decreased series resistance, lower dark current, increased rectifying ratio, and increased parallel resistance in PSCs device, which brought a sharp increase in FF from 32.5 to 55.1%. Surface passivation brought slight increase of $J_{SC}$ than control devices. The improvement in all key parameters of PSCs contributed the significant increasement in PCE. Statistical data of 10 samples with and without MAI treatment were shown in Fig. 6e, f, all of which showed higher $V_{OC}$, FF, $J_{SC}$, and PCE after treatment. Metals with different work functions were introduced to evaluate the cathode contribution, such as Au, Al, and Cu. There was similar device performance in $J$-$V$ curves of PSCs with Al, Cu or Au as cathode and Au as anode as shown in Supplementary Fig. 3, which did not highly dependent on the work function of cathode. It might cause by the different charge carriers transport pathways from sandwiched PSC. In regular sandwiched PSCs, work function differences in anode and cathode might lead to much higher built-in potential than that in lateral device, because the distance between two electrodes was about 50 μm in our device which was much longer than that of around 500 to 700 nm in regular PSCs.

Scalable fabrication process was also investigated by using interdigital electrodes by the same procedure as shown in Fig. 1. The lateral structure PSCs could achieve much lower series resistance than ITO or FTO based PSCs especially after scale up, because long range charge transport was realized by highly conductive metals for both anode and cathode in

lateral structured SC devices, which potentially further promoted the FF and efficiency for PSCs. However, the performance of the current large-area SC-PSCs was still limited perovskite SC itself, because it was hard to get large-area thin crystals with uniform low defects surface over large area. Nevertheless, as shown in Supplementary Fig. 4, the scaled-up device with 36 times larger active area still exhibited a PCE of 6.30% with a $J_{SC}$ of 4.87 mA cm$^{-2}$, a FF of 42% and a $V_{OC}$ of 0.77 V, which provide an effective strategy to scale up lateral-structure SC-PSCs device.

Besides the efficiency, the stability was also a main problem for PSCs and there were still few works reported long-term operation stability of SC-PSCs. Here, our SC-PSC as continuously operated at MPP condition with 1 Sun illumination in glovebox without cooling stage and the photocurrent output was tracked to evaluate the operational stability. The device performance variation in 200 h continuous operation was shown in Fig. 6g. Inspiringly, our champion device still maintains 99.77%, in range of variation caused by environment such as temperature, of its initial efficiency without observed degradation after operation at MPP 1 Sun condition for 200 h. It should be noted that our devices were based on MAPbI$_3$ composition, which was usually considered less stable than mixed cation perovskite while it was still hard to grow mixed cation perovskite SCs at present, however, the SC-PSCs still exhibited ultra-high stability. The high stability of SC-PSCs is critical for the development of PSCs including both of SC and PC devices, which provide a potential

pathway to further promoting the overall stability of PSCs by crystallinity engineering.

## Discussion

In conclusion, a kind of stable and efficient lateral structure PSCs is fabricated based on MAPbI$_3$ single crystal by a simple MAI treatment procedure. The MAI treatment significantly passivates surface defects, enhances surface conductivity and promotes the efficiency of lateral structure SC-PSC. Excellent long-term operation stability of single crystal perovskite solar cell is verified with no degradation after 200 h continuous operation at MPP 1 Sun condition. With the development of large-area thin single crystals growth and surface passivation technique, it will show a bright future and potentials towards efficient perovskite mono-crystalline solar cells with dramatically reduce material and fabrication cost.

## Methods

**Fabrication of lateral-structure MAPbI$_3$ single-crystal device**. Thin single crystals were used for device fabrication which was grown by space-confined method[16,51,52]. For MAI treatment, 0.5 mg mL$^{-1}$ MAI solution in IPA was deposited on the single crystal by spin coating at 3000 r.p.m and was dried at 50 °C for 10 min on a hot plate. Then, about 65 nm Au was thermally evaporated on the top of half part of the single crystal as anode, and the other layers of 20 nm C60 and 7.5 nm BCP were deposited on the other part of the single crystal. Subsequently, a gap width of 50 μm and length of 1 mm shadow mask was covered on the interfacial edge of these two areas. Finally, another 65 nm Au (Al or Cu)was thermal deposited on the top surface as the cathode. The preparation of large-area devices was similar just changing the electrodes to interdigital electrode.

**Material and device characterization**. J-V measurements on lateral-structure PSCs were conducted in a probe station chamber under white light (25 mW cm$^{-2}$) through a quartz window and the 1 Sun data were evaluated by using a sun simulated AM 1.5G irradiation with AAA class solar simulator (SOLARBEAM-02-3A, CROWNTECH.INC.). The light intensity of 100 mW cm$^{-2}$ was calibrated by standard Si photodiode detector (SRC-1000-TC-K-QZ-C, VLSI Standards). The single device area was 50 μm × 1 mm and the 36 times scale up device area was 1.85 mm$^2$. A metal photomask was used during measurement to cover all the exposed crystal surface except working area and electrodes, while the working area was defined as the channel area between the metal electrodes. The PCE determination normalized by the active area for the interdigitated-structure devices. Two probes were used to contact each electrode on top surface of the device. The J-V curves of lateral MAPbI$_3$ single-crystal solar cell devices were measured by a Keithley 2400 source meter, and the dark current density-voltage curves of the devices were tested in the same way as the photocurrent density-voltage curve test method except in complete darkness. The XRD spectra were obtained using the X-ray diffractometer (Empyrean) equipped with a Cu tube operated at 40 kV and 30 mA. The views of the single crystal surface were collected with a field emission scanning electron microscopy (HITACHI, SU8020). Time-resolved fluorescence spectra were measured by a FLS920 time-corrected single photon counting system. The KPFM measurement was carried out by a Bruker FastScan AFM system. The XPS data were obtained by ESCALAB 250Xi X-ray photoelectron spectrometer. TOF-SIMS study was conducted by the IONTOF V. The FTIR spectroscopy was gained by VERTEX 80 V (Brucker).

**Reporting summary**. Further information on experimental design is available in the Nature Research Reporting Summary linked to this paper.

## Data availability

The data that support the plots within this paper are available from the corresponding author upon request. The source data underlying Figs. 3a–g, 4c–d, 5a–c, and 6a–g and Supplementary Figs. 1–4 are provided as a Source Data file.

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

## Acknowledgements

This work was supported by the National Natural Science Foundation of China. (No. 21875089).

## Author contributions

Q.D. conceived the idea, supervised the project and conduct the initial experiment; Y.S. and W.B. grew the single crystals, Y.S. fabricated the devices, conducted the characterization and do most of experiments. Y.K., A.W., and X.L. contributed to the SEM, XRD, and TRPL characterization. Q.D. and Y.S. wrote the paper.

## Competing interests

The authors declare no competing interests.
