## [Peer Review File · Nature Communications]

Reviewers' comments:

Reviewer #1 (Remarks to the Author):

This work describes perovskite solar cell employing a lateral device structure based on single crystal MAPbI₃. Surface treatment of single crystal with MAI showed better performance than without MAI treatment. This improvement was due to the reduced trap-mediated recombination associated with surface treatment. It is high efficiency with a lateral structure, but except for this specific device structure little scientific approach can be found from the current work. Thus this work may be appropriate for specific journal dealing with perovskite solar cells.

Reviewer #2 (Remarks to the Author):

In the present paper, the authors fabricated a single crystal MAPbI₃ based solar cell with a lateral structure. They show that the performance can be significantly enhanced when passivating with MAI. In addition, they found the devices to be relatively stable after 1000 hours of operation at MPP. The authors also show a higher power conversion efficiency than previously reported for the same device architecture (from 5.9% to 12.27%). As suggested in the manuscript, optimizing lateral structures is indeed important, since later lateral structures display industrial advantages absent in vertical types. However, it is also important to be cautious when reporting such high efficiencies and avoid overestimation of the photovoltaic parameters. Thus, the following points are suggested:

1- Was the stability measured at 1 sun or 0.25 sun? If the latter, it would be better to mention that the efficiency and stability were measured under 0.25 sun in the Abstract to avoid confusion.

2- Is it possible to acquire an EQE spectrum to show that there's no overestimation in the short-circuit current? The short-circuit current seems to be a bit high from a device consisting of only one transporting layer (ETL).

3- The concept of using MAI to passivate single crystals is not very novel, but has been shown to increase performance in vertical solar cells: Chen, Z. et al. Thin single crystal perovskite solar cells to harvest below-bandgap light absorption. Nature Communications 8, (2017).

The authors should mention that it was found to be an effective way to enhance performance, and then say that they explored it for lateral solar cells.

Reviewer #3 (Remarks to the Author):

In this work, Song et al. fabricated MAPbI₃ single crystal (SC) based lateral-structured perovskite solar cells employing gold as anode deposited directly on the MAPbI₃ SC and C60/BCP/Au as cathode. In addition, MAI treatment led further to device improvement attaining the highest PCE of 12.27% (under 0.25 sun). Overall this work is of interest for the community working with perovskite solar cells. However, there are several points that need to be clarified, which are noted below.

The work by Chen et al. ACS Energy Lett. (2019) 1258-1259 employs MAPbI₃ single crystal (thickness = 20 microns) with sandwich structure showing an outstanding PCE of 21.09% and FF up to 84.3%. Although authors criticized about the use of expensive ITO, this work should be described in the Introduction section stating the above PCE and FF can be achieved in MAPbI₃ single crystals. Authors

are also invited to provide a better clarification of the advantages employing the lateral-structure in addition to disadvantage of ITO. This work will be more convincing if authors propose further strategies that can help enhance PCE.

Authors employed an ultra-thin layer of MAI promoting a better matching of MAPbI₃ single crystal energy level with the adjacent gold electrode Fermi level. However, the mechanisms how MAI interacts with MAPbI₃ surface was not discussed. Authors performed SEM and XRD showing that morphology and crystallinity had minimum influences with MAI treatment. Discussion of chemical composition analysis is equally important in this work to provide readers further better understanding. I would like to suggest the authors to perform XPS, FTIR, and/or TOF-SIMS measurements for attaining further chemical composition analysis. The reference [J. Phys. Chem. Lett. 8 (2017) 3947-3953] may be of help.

Authors discussed that MAI treatment leads to suppression of surface trap densities observed by blue-shift in PL and longer lifetimes in TRPL measurements. It will reinforce authors' statement of surface trap density healing if another technique is also employed, for example, dark I-V for determining the trap-filled limit voltage. The references [Opt. Express 26 (2018) 26307-26316 (Table 2); ACS Energy Lett. 4 (2019) 779-785] are further suggested. Discussion on the correlation between chemical composition analysis and defect healing will help readers to understand better how MAI leads to performance improvements.

I'm wondering about the solar cell performance if a low work function metal of aluminum is used as cathode on the ETL (C60/BCP). Please provide this information in the manuscript. The proposed method by the authors has the advantage that different electrode materials can be deposited (i.e., asymmetrical electrodes). If authors take advantage of work function differences in anode and cathode, it may lead to even higher PCEs due to built-in potential.

Because authors employed copper as reference electrode for KPFM measurements, I'm wondering if they tried single crystal solar cell performance testing with copper electrodes and compare with that of employing gold electrodes. Table 1 could be further extended with these solar cell parameters (J_{sc} , V_{oc} , FF, PCE).

Clarification of the PCE determination normalized by the active area is needed as interdigitated-structure is employed. How the active area was determined in this work? Please incorporate in Methods section. Please state explicitly the active area size (50 microns x 1 mm ?). It is mentioned that "scaled-up device with 36 times larger active area" correspond to Fig. 6E. Please provide the absolute total active area size?

As a minor comment, a critical checking for small spelling and grammar inaccuracies is recommended. For example, "significant increasement in PCE"; "In order to further confirmed..."; "current large area SC-PSCs was mainly limited perovskite SC itself";

Reviewer #4 (Remarks to the Author):

In this paper, Efficient Lateral-structure Perovskite Single Crystal Solar Cells with High Operational Stability by Surface Doping, the author's create IBC architecture solar cells from single crystals of MAPbI₃.

I do not recommend publication of this article for several reasons.

1. Neither the architecture, the material (single crystals), or the results are new to the field and the

publication reveals no new information, no new science, and the results are not record efficiencies. 2. The authors claim a doping effect based on a single characterization, KPFM. Shifts in the surface potential can be caused by a variety of factors such as a thin surface layer of a different material, dipole alignment, ferroelectricity or charge screening due to ion motion; all of these have been previously reported for this material and this report provides no information about which effect is present here.

3. Most notably, the authors report the solar cell results at an intensity of 0.25 suns. I can't believe that the authors never tested these at 1 sun and those would be the relevant results. If they didn't then it's simply irrelevant science. It is known that this material's performance and stability are effected by coupled processes which are induced by light, heat, electric field, moisture and oxygen. Performing tests in vacuum and at low illumination removes two of the principal perturbations which cause degradation and low performance.

The use of 0.25 suns illumination, as well as the lack of any substantial characterization of the electronic effects of the MAI, appears to be an attempt by the authors to get a "record efficiency" device published. It is a sensationalist presentation of the data and should not be published.

Reviewer #5 (Remarks to the Author):

25/08/2019

Review of the manuscript entitled "Efficient Lateral-structure Perovskite Single Crystal Solar Cells with High Operational Stability by Surface Doping" by Song et al., submitted to Nature Communications.

The manuscript reports on efficient and stable back-contact devices based on single crystal MAPbI₃ perovskites. Devices with efficiencies of up to 12.3% and operational stability over 1000 hrs are demonstrated. The authors show that by doping the perovskite film with MAI an enhanced energy level alignment with the anode is achieved, resulting in improved device efficiencies. The authors also attribute the improved device performance to a well passivated and highly conductive perovskite surface, which in turn leads to efficient charge collection.

This work is of definite interest to the perovskite community, specifically to those working on single crystals and back-contact structures. Throughout the manuscript, the authors present a thorough comparison between doped and un-doped devices. The main findings are clearly explained and the suggested methods for scalable device fabrication are remarkable.

However, I find some of the discussion a bit difficult to read and understand. Therefore, I suggest the following revisions:

In the introduction (first paragraph), the authors mention wrong values of the efficiencies of the very first and latest reported perovskite solar cells. The correct values are 3.81% (not 3.9%, see Table 1, Kojima et al. 2009) and 25.2% (not 24.2%, please check the latest NREL chart, <https://www.nrel.gov/pv/assets/pdfs/best-research-cell-efficiencies.20190802.pdf>).

In the introduction (second paragraph), the authors cite works on silicon solar cells employing the back-contact structure. Since reference 28 (Tainter et al.) is on perovskites, I think it should be removed from this sentence.

Another sentence must be added citing the following works on back-contact perovskite solar cells:

- Richard H. Friend, Felix Deschler, Luis M. Pazos-Outón, Mojtaba Abdi-Jalebi, Mejd Alsari, Back-Contact Perovskite Solar Cells. *Scientific Video Protocols*, 1, 1, (2019), <https://doi.org/10.32386/scivpro.000005>, which provides an overview of back-contact perovskite solar cells;
- A. N. Jumabekov, E. Della Gaspera, Z. Q. Xu, A. S. R. Chesman, J. van Embden, S. A. Bonke, Q. Bao, D. Vak & U. Bach. Back-contacted hybrid organic-inorganic perovskite solar cells. *Journal of Materials Chemistry C* 4, 3125-3130, (2016). <https://doi.org/10.1039/C6TC00681G>
- Q. Hou, D. Bacal, A. N. Jumabekov, W. Li, Z. Wang, X. Lin, S. H. Ng, B. Tan, Q. Bao, A. S. R. Chesman, Y.-B. Cheng & U. Bach. Back-contact perovskite solar cells with honeycomb-like charge collecting electrodes. *Nano Energy* 50, 710-716, (2018). <https://doi.org/10.1016/j.nanoen.2018.06.006>
- Z. Hu, G. Kapil, H. Shimazaki, S. S. Pandey, T. Ma & S. Hayase. Transparent Conductive Oxide Layer and Hole Selective Layer Free Back-Contacted Hybrid Perovskite Solar Cell. *The Journal of Physical Chemistry C* 121, 4214-4219, (2017). <https://doi.org/10.1021/acs.jpcc.7b00760>
- A. N. Jumabekov, J. A. Lloyd, D. M. Bacal, U. Bach & A. S. R. Chesman. Fabrication of Back-Contact Electrodes Using Modified Natural Lithography. *ACS Applied Energy Materials* 1, 1077-1082, (2018). <https://doi.org/10.1021/acsaem.7b00213>
- Mejd Alsari, Oier Bikondo, James Bishop, Mojtaba Abdi-Jalebi, et al. In situ simultaneous photovoltaic and structural evolution of perovskite solar cells during film formation. *Energy & Environmental Science*, 11, 383, (2018), <https://doi.org/10.1039/C7EE03013D>
- Your ref. 28, Tainter et al. <https://doi.org/10.1016/j.joule.2019.03.010>

In the introduction (second paragraph), the sentence "The lateral structure was also a stand- free structure which did not require expensive indium tin oxide (ITO) substrate and did not have interfacial stability issues as that in traditional sandwiched-structure PSCs." does not link very well with the previous sentence and therefore needs rewriting.

In the results and discussions, the size of the back-contact devices is not clear from the discussion. I suggest the authors to show the sizes (electrodes spacing and electrodes sizes) in the schematics presented in Figure 1. I also suggest the authors to add another figure in Figure 1 where they show a schematic of the two-electrode devices to further clarify the difference between the small-area and large-area devices.

Table 1, an average of two values doesn't make statistical sense (columns SC-Cu and MAI/SC-Cu); please add the same amount of measurements used in the first two columns (SC-Au and MAI/SC-Au).

Please revise XRD indexing in Figure 4 (check <https://pubs.rsc.org/en/content/articlelanding/2016/NJ/C6NJ00188B#!divAbstract>) and/or provide references.

The authors mention 'The SC-PSC devices showed nearly 50 times increase in rectifying ratio'. Wouldn't it be clearer if the authors mention just the FF? As an alternative can they comment on how this rectifying ratio is calculated?

The authors do not mention the active area of the solar cells as defined by the photomask.

In the results and discussions, the sentence "In ideal condition, the device performance would not be limited by the resistance of electrodes when scaled up, because almost all surface of the SCs were covered by metal electrode which would not significantly increase the series resistance of SC-PSCs compared with small area devices, which was an advantage of lateral-structure

SC-PSCs compared with current sandwich structured PSCs and most of thin film solar cells." is too long and hard to read and therefore needs rewriting.

Overall language/sentences throughout the manuscript need revising for better readability.

Response letter to Reviewers' comments:

Reviewer #1 (Remarks to the Author):

This work describes perovskite solar cell employing a lateral device structure based on single crystal MAPbI₃. Surface treatment of single crystal with MAI showed better performance than without MAI treatment. This improvement was due to the reduced trap-mediated recombination associated with surface treatment. It is high efficiency with a lateral structure, but except for this specific device structure little scientific approach can be found from the current work. Thus this work may be appropriate for specific journal dealing with perovskite solar cells.

Response: Thanks for the recognition of the high efficiency of perovskite solar cell (PSC) with a lateral structure in this work!

This work promoted efficiency, stability and device area of lateral structured PSC to a comparable level to that of regular solar cells, such as organic solar cells, quantum dots solar cells as well as regular perovskite solar cells, which provide another potential pathway for efficient photovoltaic with really low cost benefit to its ITO free device structure. It was the first time to report the operational stability of single crystal perovskite device, and the encouraging stability result showed *pure MAPbI₃* composition can be very stable in operational condition without further bulk doping, or additional functional buffer layers, which was always believed to have low intrinsic stability. The breakthrough in this work will help researchers to further understand the intrinsic stability of perovskite materials and devices which is important for the whole perovskite research community instead of a specific direction.

In order to reinforce the scientific approach in this work as mentioned in the comment, we revised the paper by adding more deep analysis of the doping process, which was the key factor to realize such efficient charge transport across tens of microns distance, which had never been realized in previously reported perovskite solar cells. We take X-ray photoelectron spectroscopy (XPS), Fourier transform infrared (FTIR) spectroscopy as well as time-of-flight secondary ion mass spectroscopy (TOF-SIMS) measurement to show how MAI interact with MAPbI₃ surface, and evaluate the benefits of MAI doping by qualifying the surface trap density and carrier mobility change before and after MAI doping with space-charge-limited current (SCLC) method. The light intensity dependent *J-V* properties was also shown for deep understanding of the **recombination mechanism**. All the discussion and changes were marked in the revised manuscript for reconsideration. It is evident that MAI doping alone contributed to significant enhancement in both surface electronic structure as well as charge transport properties, resulted in highly stable and efficient PSCs even when electrode spacing was as large as 50 microns.

We revised our manuscript as below in **Page 9, 11, 13, 14 and 14**:

(1) XPS, FTIR and TOF-SIMS measurements were taken for attaining further chemical composition analysis to show how MAI interact with MAPbI₃ surface:

Page 9 Line 158: “XPS [*J Phys Chem Lett* **8**, 3947-3953 (2017)], FTIR [*Journal of Materials Chemistry A* **4**, 17018-17024 (2016)] and TOF-SIMS [*ACS Appl Mater Interfaces* **11**, 30911-30918 (2019); *Nat Energy* **3**, 68-74 (2018)] measurements were taken for attaining further chemical composition information to show how MAI interact with MAPbI₃ surface as shown in Fig.3C-G. The metallic lead was detected on surface of as grown MAPbI₃ single crystal (SC) as

shown in XPS result, which was believed as a main factor of surface traps. After MAI doping, the peak from metallic lead was effectively suppressed and a weak MAI peak was emerging in XPS spectrum, which was also evidenced by the increased FTIR response around 3400 cm^{-1} directly related to N–H stretching vibrations for the MAI. TOF-SIMS measurement was also taken to reveal the surface and bulk properties of as grown and MAI doped MAPbI₃ SCs in detail. The MAI-perovskite interaction was mainly a surface reaction instead of bulk doping evidenced by the TOF-SIMS study for both as grown and MAI treated SCs. The lower crystallized SC surface, with higher initial second ion intensity, was found mainly on the top 5 nm near surface of SCs evidenced by MAI and PbI₂ related signals in TOF-SIMS results as shown in Fig. 3G. Based on the XPS, FTIR and TOF-SIMS results, the very thin lower crystallized surface of SCs within 5 nanometers with metallic lead was detected and believed as a main origin of surface traps, MAI treatment contributed a surface doping and effectively passivated SC surface by suppressed formation of metallic lead.”

XPS

FTIR

(2) Investigation on the effect of MAI doping by qualifying the surface trap density and carrier mobility change by SCLC method:

Page 11 Line 193: “The MAI doping contributed significantly to the decreased surface trap density and increased carrier mobility across the surface of the $\text{CH}_3\text{NH}_3\text{PbI}_3$ SCs, which were evaluated by space-charge-limited current (SCLC) measurements [*Opt Express* **26**, 26307-26316 (2018); *ACS Energy Lett* **4**, 779-785 (2019)] as shown in Fig. 4D. For SCLC method in lateral devices, a gap-type device was fabricated using thin single crystal with C60/BCP layers on top and Au electrodes with channel width of 50 μm . [*Nat Commun* **8**, 15882 (2017); *Applied Physics Letters* **11**, 183-185 (1967)] The dark current–voltage (J - V) curve of the electron-only SC device was shown in Figure 4C. From this J - V curve, we could obtain the carrier mobility for gap-type devices by the Geurst’s SCLC model [*Nat Commun* **8**, 15882 (2017); *App Phys Lett* **11**, 183-185 (1967)].

$$I = \frac{2\mu\epsilon_0\epsilon_r W}{\pi} \left(\frac{V^2}{L^2}\right) \quad (1)$$

According to the Geurst theory the threshold voltage for a gap-type structure was equal to [*Appl Phys Lett* **11**, 183-185 (1967)]

$$V_T = \frac{\pi\sigma_0 L}{4\epsilon_0\epsilon_r} \quad (2)$$

where μ is the carrier mobility, σ_0 is the surface charge density per unit area, ϵ_0 is the vacuum permittivity and ϵ_r is relative dielectric constant of perovskite, 32 [*Science* **347**, 967-970 (2015)]. W and L are the device width and interelectrode distance, respectively.

After MAI doping treatment, the electron mobility of the SC was effectively enhanced to $1.16 \text{ cm}^2 \text{ V}^{-1} \text{ s}^{-1}$ from that of $0.60 \text{ cm}^2 \text{ V}^{-1} \text{ s}^{-1}$ in SC without treatment. The surface trap density was also significantly suppressed from $6.67 \times 10^9 \text{ cm}^{-2}$ to $4.51 \times 10^9 \text{ cm}^{-2}$ after MAI doping.”

(3) Page 13 Line 229: “Intensity dependence of J_{SC} and V_{OC} curves was shown in Fig. 5C and D with light intensity in range of 0.0005 to 1.5 Sun. The ideal factor (n) was a vital indicator that reflected the charge recombination in working devices. [*Adv Mater* **27**, 1837-1841 (2015)] The n could be calculated by the intensity dependence of V_{OC} curves, and the curves were fitted using the following equation [*Science* **358**, 768-771 (2017)]:

$$V_{oc} = \frac{nKT}{q} \ln\left(\frac{J_{sc}}{J_0}\right)$$

By calculation, n equaled to 2.3 (doping with MAI) and 3.1 (control), respectively. For control device, the higher value of n was gained that was attributed to the presence of severer nonradiative trap-assisted recombination. [*Adv Energy Mater* **8**, (2018)] And this proved that MAI had a marvelous passivation effect.”

Page 14 Line 246: “Light intensity dependent J - V curves of lateral structured SC-PSCs under AM1.5G illumination with different light intensity from 0.05 to 1.5 Sun by an AAA grade solar simulator, which was a standard test condition for regular PSCs, was shown in Fig. 6A-B. The PSC based on as grown SCs showed much faster degradation rate in photocurrent density under higher light intensity (Fig. 5C). As a strong contrast, PSC with MAI doping kept liner relationship between photocurrent density and light intensity and a PCE of 11.52 % was achieved under 1 Sun condition and a champion PCE of 12.76 % under 0.5 Sun benefited from the suppressed surface recombination.”

Page 15 Line 257: “It is also worth to be noted that our crystal growth as well as device fabrication was processed in air condition. There is no much difference for J - V curves of devices tested in vacuum or in air. (Fig. S1)”

Reviewer #2 (Remarks to the Author):

In the present paper, the authors fabricated a single crystal MAPbI₃ based solar cell with a lateral structure. They show that the performance can be significantly enhanced when passivating with MAI. In addition, they found the devices to be relatively stable after 1000 hours of operation at MPP. The authors also show a higher power conversion efficiency than previously reported for the same device architecture (from 5.9% to 12.27%).

Response: Thanks for the recognition of the achievement in the stability and enhanced efficiency in this work!

As suggested in the manuscript, optimizing lateral structures is indeed important, since later lateral structures display industrial advantages absent in vertical types. However, it is also important to be cautious when reporting such high efficiencies and avoid overestimation of the photovoltaic parameters. Thus, the following points are suggested:

Comment 1- Was the stability measured at 1 sun or 0.25 sun? If the latter, it would be better to mention that the efficiency and stability were measured under 0.25 sun in the Abstract to avoid confusion.

Response to C1: Thanks for the comments, we have shown the *J-V* curves and power conversion efficiency (PCE) of our device at different light intensities from 0.05 to 1.5 Sun and clarified the stability test condition in the revised **Abstract** and also in introduction part as:

Abstract: “By optimizing anode contact with a simple surface doping treatment, the open circuit voltage and fill factor dramatically increased and promoted the PCE of the devices exceeding 11 % (0.05 to 1 Sun) compared to that of 5.9 % (0.25 Sun) of the best lateral-structure SC-PSCs previously reported. Devices show excellent operational stability and maintained 93 % of its initial PCE after 1000 hours’ continuous operation at maximum power point condition (0.25 Sun).”

Page 5 Line 87: “A PCE exceeding 11 % achieved at different light intensity from 0.05 to 1 Sun in MAI doped PSCs compared to that of 5.9 % (0.25 Sun) of the best lateral-structure SC-PSCs previously reported.”

Comment 2- Is it possible to acquire an EQE spectrum to show that there's no overestimation in the short-circuit current? The short-circuit current seems to be a bit high from a device consisting of only one transporting layer (ETL).

Response to C2: Thanks for the comment. It is still challenging to get absolute external quantum efficiency (EQE) spectrum to evaluate the photo-response properties of lateral single crystal device, because the photo-generated charges would transport across at least tens of micrometers across surface of single crystal to be collected by electrodes, which was strongly affected by the surface traps especially under weak illumination during EQE measurement. And the photo-response of lateral device was also affected by the large resistance of perovskite under weak light which make the surface in “tune off” condition. The light intensity dependent photo-response for lateral structured single crystal perovskite solar cells (SC PSCs) was shown below, with significantly decreased device performance in range of weak light intensity comparable to that of EQE measurement (0.0001 Sun), and the photocurrent is even hard to be measured at a stable value.

The enhanced PCE of our device compared with previous reported work was mainly contributed by the enhanced V_{OC} and FF, while the J_{SC} was merely increased slightly. In order to avoid overestimation of the photovoltaic parameters, we also show the $J-V$ curves of our device under different light intensity from 0.05 to 1.5 Sun under AM1.5 condition by using AAA class solar simulator, which make our devices comparable to regular vertical structure perovskite solar cells. The PCE is still similar under 1 Sun AM1.5 condition.

Nevertheless, we still try our best to get EQE response curve below to show the photo response property of lateral PSCs. Obviously, there was a significant red shift around 50 nm for EQE cutoff by the sub-bandgap absorption of single crystal MAPbI₃. The wide absorption range as well as higher absorption efficiency were benefited from the direct light absorption of our ITO free device structure without light loss by transparent electrode in this lateral device structure, which possesses a higher potential maximum photocurrent than regular ITO or FTO based polycrystalline PSCs including hole transport layer (HTL) free PSCs.

Comment 3- *The concept of using MAI to passivate single crystals is not very novel, but has been shown to increase performance in vertical solar cells: Chen, Z. et al. Thin single crystal perovskite solar cells to harvest below-bandgap light absorption. Nature Communications 8, (2017).*

The authors should mention that it was found to be an effective way to enhance performance, and then say that they explored it for lateral solar cells.

Response to C3: Thanks for the suggestion. We highlighted the previous work in the revised paper and added discussion as: “which was found to be an effective way to enhance performance due to the passivation effect and here we explored it for lateral solar cells.”

Reviewer #3 (Remarks to the Author):

In this work, Song et al. fabricated MAPbI₃ single crystal (SC) based lateral-structured perovskite solar cells employing gold as anode deposited directly on the MAPbI₃ SC and C60/BCP/Au as cathode. In addition, MAI treatment led further to device improvement attaining the highest PCE of 12.27% (under 0.25 sun). Overall this work is of interest for the community working with perovskite solar cells. However, there are several points that need to be clarified, which are noted below.

Response: Thanks for the recognition of the significance of this work!

Comment 1: (1) The work by Chen et al. ACS Energy Lett. (2019) 1258-1259 employs MAPbI₃ single crystal (thickness = 20 microns) with sandwich structure showing an outstanding PCE of 21.09% and FF up to 84.3%. Although authors criticized about the use of expensive ITO, this work should be described in the Introduction section stating the above PCE and FF can be achieved in MAPbI₃ single crystals.

Response to C1: Thanks for the recognition of the advantage of single crystal perovskite solar cells (SC-PSCs) with a lateral device structure. We have further highlighted this excellent work in the revised paper as below:

Page 4 Line 66: “Although the best SC-PSC is still based on sandwich structure with expensive ITO, an outstanding power conversion efficiency (PCE) of 21.09 % and fill factor (FF) up to 84.3 % was achieved in a 20 microns thick MAPbI₃ single crystals device, which show huge potential for developing high performance PSCs based on perovskite single crystal.”

Comment 2: Authors are also invited to provide a better clarification of the advantages employing the lateral-structure in addition to disadvantage of ITO. This work will be more convincing if authors propose further strategies that can help enhance PCE.

Response to C2: We further clarified advantages of lateral structure solar cells as below:

Page 3 Line 61: “The lateral structured SC-PSCs could further promote the light absorption efficiency because light could directly illuminate on crystal surface without losing caused by absorption of glass substrate as well as conductive electrode, which would contribute to the enhanced photocurrent and efficiency than regular PSC based on ITO or fluorine-doped tin oxide (FTO) electrode”.

Page 17 Line 229: “The lateral structure PSCs could achieve much lower series resistance than ITO or FTO based PSCs especially after scale up, because long range charge transport was realized by highly conductive metals for both anode and cathode in lateral structured SC devices, which potentially further promoted the fill factor and efficiency for PSCs”.

Page 3 Line 56: “Stand-free lateral structure SC-PSC immune from strain stress induced mechanical device degradation due to different thermal expansion coefficients for multiple functional layers as well as substrates, which potentially increased the device stability than regular sandwich structures”.

Comment 3: Authors employed an ultra-thin layer of MAI promoting a better matching of MAPbI₃ single crystal energy level with the adjacent gold electrode Fermi level. However, the mechanisms how MAI interacts with MAPbI₃ surface was not discussed. Authors performed SEM and XRD showing that morphology and crystallinity had minimum influences with MAI treatment. Discussion of chemical composition analysis is equally important in this work to provide readers further better understanding. I would like to suggest the authors to perform XPS, FTIR, and/or TOF-SIMS measurements for attaining further chemical composition analysis. The reference [*J. Phys. Chem. Lett.* 8 (2017) 3947-3953] may be of help.

Response to C3: Thanks for the helpful suggestion! As suggested, we take **X-ray photoelectron spectroscopy (XPS)**, Fourier transform infrared (**FTIR**) spectroscopy and time-of-flight secondary ion mass spectroscopy (**TOF-SIMS**) measurement for attaining further chemical composition analysis to show how MAI interact with MAPbI₃ surface. Related discussion was added in revised manuscript as below:

(1) XPS, FTIR and TOF-SIMS measurement for attaining further chemical composition analysis to show how MAI interact with MAPbI₃ surface:

Page 9 Line 158: “XPS [*J Phys Chem Lett* 8, 3947-3953 (2017)], FTIR [*J Mater Chem A* 4, 17018-17024 (2016)] and TOF-SIMS [*ACS Appl Mater Interfaces* 11, 30911-30918 (2019); *Nat Energy* 3, 68-74 (2018)] measurements were taken for attaining further chemical composition information to show how MAI interact with MAPbI₃ surface as shown in Fig.3C-G. The metallic lead was detected on surface of as grown MAPbI₃ single crystal (SC) as shown in XPS result, which was believed as a main factor of surface traps. After MAI doping, the peak from metallic lead was effectively suppressed and a weak MAI peak was emerging in XPS spectrum, which was also evidenced by the increased FTIR response around 3400 cm⁻¹ directly related to N–H stretching vibrations for the MAI. TOF-SIMS measurement was also taken to reveal the surface and bulk properties of as grown and MAI doped MAPbI₃ SCs in detail. The MAI-perovskite interaction was mainly a surface reaction instead of bulk doping evidenced by the TOF-SIMS study for both as grown and MAI treated SCs. The lower crystalized SC surface, with higher initial second ion intensity, was found mainly on the top 5 nm near surface of SCs evidenced by MAI and PbI₂ related signals in TOF-SIMS results as shown in Fig. 3G. Based on the XPS, FTIR and TOF-SIMS results, the very thin lower crystalized surface of SCs within 5 nanometers with metallic lead was detected and believed as a main origin of surface traps, MAI treatment contributed a surface doping and effectively passivated SC surface by suppressed formation of metallic lead”

XPS

FTIR

TOF-SIMS

Comment 4: Authors discussed that MAI treatment leads to suppression of surface trap densities observed by blue-shift in PL and longer lifetimes in TRPL measurements. It will reinforce authors' statement of surface trap density healing if another technique is also employed, for example, dark

I-V for determining the trap-filled limit voltage. The references [*Opt. Express* 26 (2018) 26307-26316 (Table 2); *ACS Energy Lett.* 4 (2019) 779-785] are further suggested. Discussion on the correlation between chemical composition analysis and defect healing will help readers to understand better how MAI leads to performance improvements.

Response to C4: Thanks for the helpful comments and suggested references. As suggested, dark *J-V* for determining the trap-filled limit voltage was measured and there was a significantly decreased trap density by MAI doping, which reinforces our statement again.

We add discussion for further analyzing the correlation between chemical composition analysis and defect healing:

Page 11 Line 193: “The MAI doping contributed significantly to the decreased surface trap density and increased carrier mobility across the surface of the $\text{CH}_3\text{NH}_3\text{PbI}_3$ SCs, which were evaluated by space-charge-limited current (SCLC) measurements [*Opt Express* 26, 26307-26316 (2018); *ACS Energy Lett* 4, 779-785 (2019)] as shown in Fig. 4D. For SCLC method in lateral devices, a gap-type device was fabricated using thin single crystal with C60/BCP layers on top and Au electrodes with channel width of 50 μm . [*Nat Commun* 8, 15882 (2017); *Appl Phys Lett* 11, 183-185 (1967)] The dark current–voltage (*J-V*) curve of the electron-only SC device was shown in Figure 4C. From this *J-V* curve, we could obtain the carrier mobility for gap-type devices by the Geurst’s SCLC model [*Nat Commun* 8, 15882 (2017); *App Phys Lett* 11, 183-185 (1967)].

$$I = \frac{2\mu\epsilon_0\epsilon_r W}{\pi} \left(\frac{V^2}{L^2}\right) \quad (1)$$

According to the Geurst theory the threshold voltage for a gap-type structure was equal to [*Appl Phys Lett* 11, 183-185 (1967)]

$$V_T = \frac{\pi\sigma_0 L}{4\epsilon_0\epsilon_r} \quad (2)$$

where μ is the carrier mobility, σ_0 is the surface charge density per unit area, ϵ_0 is the vacuum permittivity and ϵ_r is relative dielectric constant of perovskite, 32 [*Science* 347, 967-970 (2015)]. W and L are the device width and interelectrode distance, respectively.

After MAI doping treatment, the electron mobility of the SC was effectively enhanced to $1.16 \text{ cm}^2 \text{ V}^{-1} \text{ s}^{-1}$ from that of $0.60 \text{ cm}^2 \text{ V}^{-1} \text{ s}^{-1}$ in SC without treatment. The surface trap density was also significantly suppressed from $6.67 \times 10^9 \text{ cm}^{-2}$ to $4.51 \times 10^9 \text{ cm}^{-2}$ after MAI doping.”

Comment 5: I'm wondering about the solar cell performance if a low work function metal of aluminum is used as cathode on the ETL (C60/BCP). Please provide this information in the manuscript. The proposed method by the authors has the advantage that different electrode materials can be deposited (i.e., asymmetrical electrodes). If authors take advantage of work function differences in anode and cathode, it may lead to even higher PCEs due to built-in potential.

Response to C5: The comments was very helpful and we indeed have optimized metals with different work functions to evaluate the cathode contribution, such as Au, Al and Cu. PSCs with all these metals can achieve optimized device performance as shown below. Discussion was added in Page 16 Line 279 as: "Metals with different work functions were introduced to evaluate the cathode contribution, such as Au, Al and Cu. There was similar device performance in *J-V* curves of PSCs with Al, Cu or Au as cathode and Au as anode as shown in Fig. S2, which did not highly dependent on the work function of cathode. It might cause by the different charge carriers transport pathways from sandwiched PSC. In regular sandwiched PSCs, work function differences in anode and cathode might lead much higher built-in potential than that in lateral device, because the distance between two electrodes was about 50 microns in our device which was much longer than that of around 500-700 nm in regular PSCs."

In this work, the reason for choosing Au as electrode is for better evaluation of the long-term stability of single crystal PSC itself without the influence of reaction or diffusion issues in Al or Cu electrode.

Comment 6: Because authors employed copper as reference electrode for KPFM measurements, I'm wondering if they tried single crystal solar cell performance testing with copper electrodes and compare with that of employing gold electrodes. Table 1 could be further extended with these solar cell parameters (J_{sc} , V_{oc} , FF, PCE).

Response to C6: As discussed above, we have added the Cu, also Al, based device performance as a new figure and make discussion in the revised paper.

Comment 7: Clarification of the PCE determination normalized by the active area is needed as interdigitated-structure is employed. How the active area was determined in this work? Please

incorporate in Methods section. Please state explicitly the active area size (50 microns x 1 mm ?). It is mentioned that “with 36 times larger active area” correspond to Fig. 6E. Please provide the absolute total active area size?

Response to C7: We have clarified: “The PCE determination normalized by the active area for the interdigitated-structure devices”. The definition of device area was added in Fig. 1E and relative discussion was added in Method section: “The single device area was 50 μm x 1 mm and the 36 times scale up device area was 1.85 mm^2 . A metal photomask was used during measurement to cover all the exposed crystal surface except working area and electrodes, while the working area was defined as the channel area between the metal electrodes.”

Comment 8: *As a minor comment, a critical checking for small spelling and grammar inaccuracies is recommended. For example, “significant increasement in PCE”; “In order to further confirmed...”; “current large area SC-PSCs was mainly limited perovskite SC itself”;*

Response to C8: Thanks for the suggestion, we have rechecked the spelling and grammar and marked the changes in blue in revised manuscript.

Reviewer #4 (Remarks to the Author):

In this paper, Efficient Lateral-structure Perovskite Single Crystal Solar Cells with High Operational Stability by Surface Doping, the author's create IBC architecture solar cells from single crystals of MAPbI₃.

I do not recommend publication of this article for several reasons.

Comment 1: *Neither the architecture, the material (single crystals), or the results are new to the field and the publication reveals no new information, no new science, and the results are not record efficiencies.*

Response to CI: Although lateral structure perovskite solar cells (PSCs) show huge potentials to realize high efficiency and low cost PSCs, there was only a few reports about lateral single crystal (SC) PSCs since the first lateral single crystal PSC was reported in 2016. There were still many challenging issues before efficient, low cost and stable SC-PSCs can be achieved. The most challenging issue was the interfacial contact and efficient charge collection for electrodes because there was nearly ten times lower build-in electric field in lateral device than that in regular PSCs. The traditional electron or hole transport layers could adjust energy level for optimized electrode contact but could not enable long range charge transport across tens microns in lateral structured PSCs and the efficiency was limited to 5.9%, which is the best reported PCE and no anode contact optimization was reported up to now. This work was the first paper to study anode contact of lateral SC-PSC by surface doping to optimize both **surface conductivity and surface potential** and promoted the PCE of lateral device exceeding 12 %.

The breakthrough in device performance of lateral device **architecture** was based on a tens microns-scale spacing between electrodes and enable really low cost fabrication, which can be achieved simply by shadow masks and does not require expensive and complicated photolithography process. At the same time, this was also the first time to realize efficient scale up device architecture for lateral structure single crystal perovskite solar cells (SC-PSCs).

We first evaluated the long-term operational stability of single crystal perovskite device, which was important for researchers working in PSCs field because this work showed **pure MAPbI₃** composition can be very stable in operation condition, which was believed to have low intrinsic stability. The encouraging stability result may help researchers to further understand the intrinsic stability of perovskite materials and devices, which is meaningful and important for the whole perovskite field.

And to the best of our knowledge, the efficiency in this work is **a record efficiency** as well as **best operational stability** that has been reported in lateral structured SC-PSCs in the literatures.

As mentioned, this work proposed surface doping could achieve both optimized surface conductivity and surface potential, which was a new **mechanism** to optimize SC-PSCs and effectively promote the PCE near double than that of the best efficiency has been reported.

In order to reinforce the scientific approach, we revised the paper by adding more deep analysis of the doping process, which was the key factor to realize such efficient charge transport across tens of microns distance, which had never been realized in previously reported perovskite solar cells. We take **X-ray photoelectron spectroscopy (XPS)**, Fourier transform infrared (**FTIR**) spectroscopy as well as time-of-flight secondary ion mass spectroscopy (**TOF-SIMS**)

measurement to show how MAI interact with MAPbI₃ surface, and evaluate the benefits of MAI doping by qualifying the surface trap density and carrier mobility change before and after MAI doping with space-charge-limited current (SCLC) method. The light intensity dependent *J-V* properties was also shown for deep understanding of the *recombination mechanism*. All the discussion and changes were marked in the revised manuscript for reconsideration. It is evident that MAI doping alone contributed to significant enhancement in both surface electronic structure as well as charge transport properties, resulted in highly stable and efficient PSCs even when electrode spacing was as large as 50 microns.

We revised our manuscript as below in **Page 9, 11, 13, 14 and 15**:

(1) XPS, FTIR and TOF-SIMS measurement was taken for attaining further chemical composition analysis to show how MAI interact with MAPbI₃ surface:

Page 9 Line 158: “XPS [*J Phys Chem Lett* **8**, 3947-3953 (2017)], FTIR [*J Mater Chem A* **4**, 17018-17024 (2016)] and TOF-SIMS [*ACS Appl Mater Interfaces* **11**, 30911-30918 (2019); *Nat Energy* **3**, 68-74 (2018)] measurements were taken for attaining further chemical composition information to show how MAI interact with MAPbI₃ surface as shown in Fig.3C-G. The metallic lead was detected on surface of as grown MAPbI₃ single crystal (SC) as shown in XPS result, which was believed as a main factor of surface traps. After MAI doping, the peak from metallic lead was effectively suppressed and a weak MAI peak was emerging in XPS spectrum, which was also evidenced by the increased FTIR response around 3400 cm⁻¹ directly related to N–H stretching vibrations for the MAI. TOF-SIMS measurement was also taken to reveal the surface and bulk properties of as grown and MAI doped MAPbI₃ SCs in detail. The MAI-perovskite interaction was mainly a surface reaction instead of bulk doping evidenced by the TOF-SIMS study for both as grown and MAI treated SCs. The lower crystallized SC surface, with higher initial second ion intensity, was found mainly on the top 5 nm near surface of SCs evidenced by MAI and PbI₂ related signals in TOF-SIMS results as shown in Fig. 3G. Based on the XPS, FTIR and TOF-SIMS results, the very thin lower crystallized surface of SCs within 5 nanometers with metallic lead was detected and believed as a main origin of surface traps, MAI treatment contributed a surface doping and effectively passivated SC surface by suppressed formation of metallic lead.”

XPS

FTIR

TOF-SIMS

(2) Investigation on the effect of MAI doping by qualifying the surface trap density and carrier mobility change by SCLC method:

Page 11 Line 193: “The MAI doping contributed significantly to the decreased surface trap density and increased carrier mobility across the surface of the $\text{CH}_3\text{NH}_3\text{PbI}_3$ SCs, which were evaluated by space-charge-limited current (SCLC) measurements [*Opt Express* **26**, 26307-26316

(2018); *ACS Energy Lett* **4**, 779-785 (2019)] as shown in Fig. 4D. For SCLC method in lateral devices, a gap-type device was fabricated using thin single crystal with C60/BCP layers on top and Au electrodes with channel width of 50 μm . [*Nat Commun* **8**, 15882 (2017); *Appl Phys Lett* **11**, 183-185 (1967)] The dark current–voltage (J - V) curve of the electron-only SC device was shown in Figure 4C. From this J - V curve, we could obtain the carrier mobility for gap-type devices by the Geurst’s SCLC model [*Nat Commun* **8**, 15882 (2017); *App Phys Lett* **11**, 183-185 (1967)].

$$I = \frac{2\mu\epsilon_0\epsilon_r W}{\pi} \left(\frac{V^2}{L^2}\right) \quad (1)$$

According to the Geurst theory the threshold voltage for a gap-type structure was equal to [*Appl Phys Lett* **11**, 183-185 (1967)]

$$V_T = \frac{\pi\sigma_0 L}{4\epsilon_0\epsilon_r} \quad (2)$$

where μ is the carrier mobility, σ_0 is the surface charge density per unit area, ϵ_0 is the vacuum permittivity and ϵ_r is relative dielectric constant of perovskite, 32[*Science* **347**, 967-970 (2015)]. W and L are the device width and interelectrode distance, respectively.

After MAI doping treatment, the electron mobility of the SC was effectively enhanced to $1.16 \text{ cm}^2 \text{ V}^{-1} \text{ s}^{-1}$ from that of $0.60 \text{ cm}^2 \text{ V}^{-1} \text{ s}^{-1}$ in SC without treatment. The surface trap density was also significantly suppressed from $6.67 \times 10^9 \text{ cm}^{-2}$ to $4.51 \times 10^9 \text{ cm}^{-2}$ after MAI doping.”

(3) Page 13 Line 229: “Intensity dependence of J_{SC} and V_{OC} curves was shown in Fig. 5C and D with light intensity in range of 0.0005 to 1.5 Sun. The ideal factor (n) was a vital indicator that reflected the charge recombination in working devices. [*Adv Mater* **27**, 1837-1841 (2015)] The n could be calculated by the intensity dependence of V_{OC} curves, and the curves were fitted using the following equation [*Science* **358**, 768-771 (2017)]:

$$V_{\text{OC}} = \frac{nKT}{q} \ln\left(\frac{J_{\text{SC}}}{J_0}\right)$$

By calculation, n equaled to 2.3 (doping with MAI) and 3.1 (control), respectively. For control device, the higher value of n was gained that was attributed to the presence of severer nonradiative trap-assisted recombination. [*Adv Energy Mater* **8**, (2018)] And this proved that MAI had a marvelous passivation effect.”

Page 14 Line 246: “Light intensity dependent J - V curves of lateral structured SC-PSCs under AM1.5G illumination with different light intensity from 0.05 to 1.5 Sun by an AAA grade solar simulator, which was a standard test condition for regular PSCs, was shown in Fig. 6A-B.

The PSC based on as grown SCs showed much faster degradation rate in photocurrent density under higher light intensity (Fig. 5C). As a strong contrast, PSC with MAI doping kept linear relationship between photocurrent density and light intensity and a PCE of 11.52 % was achieved under 1 Sun condition and a champion PCE of 12.76 % under 0.5 Sun benefited from the suppressed surface recombination.”

Page 15 Line 257: “It is also worth to be noted that our crystal growth as well as device fabrication was processed in air condition. There is no much difference for J - V curves of devices tested in vacuum or in air. (Fig. S1)”

Comment 2: The authors claim a doping effect based on a single characterization, KPFM. Shifts in the surface potential can be caused by a variety of factors such as a thin surface layer of a different material, dipole alignment, ferroelectricity or charge screening due to ion motion; all of these have been previously reported for this material and this report provides no information about which effect is present here.

Response to C2: Thanks for the suggestion. In this work, in addition to Kelvin probe force microscopy (KPFM), doping effect was also confirmed by the significantly enhanced conductivity in dark condition, which is a direct evidence for the effective doping (Page 12-13).

During KPFM test, we have already made the following actions to avoid the noise signals mentioned in the comments for accurate measurement, and we add related discussions in Page 7 Line125 in the revised manuscript: “In order to get accurate KPFM results and suppress the influence caused by surface composition, dipole moment, ferroelectricity and ion motion, all the samples for KPFM test were got from the same surface of the same piece of single crystal. Different metals of Cu and Au was deposited on the same single crystal as reference to double confirm the variation of single crystal surface potential with multiple scans at different positions.”

In order to reinforce the scientific approach and MAI doping process in this work, we revised the paper by a deep analysis to the doping process, which was the key factor to realize such

efficient charge transport across tens microns distance and never realized by any other perovskite solar cells.

As discussed above, XPS, FTIR spectroscopy as well as TOF-SIMS measurement was used to show how MAI interact with MAPbI₃ surface and evaluate the contribution of MAI doping by qualifying the surface trap density and carrier mobility change before and after MAI doping by space-charge-limited current (SCLC) method. The light intensity dependent *J-V* properties was also shown for deep understanding the *recombination mechanism*. All the discussion and changes were marked in the revised manuscript for reconsideration. It is clear to conclude simply MAI doping contributed significant enhancement in both surface electronic structure as well as charge transport properties, resulted in highly stable and efficient PSCs even when electrode spacing as large as 50 microns. Related figures and discussion were added in revised manuscript.

Comment 3: *Most notably, the authors report the solar cell results at an intensity of 0.25 suns. I can't believe that the authors never tested these at 1 sun and those would be the relevant results. If they didn't then it's simply irrelevant science. It is known that this material's performance and stability are effected by coupled processes which are induced by light, heat, electric field, moisture and oxygen. Performing tests in vacuum and at low illumination removes two of the principal perturbations which cause degradation and low performance.*

The use of 0.25 suns illumination, as well as the lack of any substantial characterization of the electronic effects of the MAI, appears to be an attempt by the authors to get a "record efficiency" device published. It is a sensationalist presentation of the data and should not be published.

Response to C3: To avoid misunderstanding and clarify the influence of test condition on device performance, here we show there is no much difference for *J-V* curves of devices tested in vacuum or in air. It is also worth to be noted that our crystal growth as well as device fabricated was processed in air condition, which is an advantage compared with the previously efficient single crystal device that has to be made in glovebox. Fig. S1 and discussion was added in Page 15 Line 257 in revised manuscript as: “It is also worth to be noted that our crystal growth as well as device fabrication was processed in air condition. There is no much difference for *J-V* curves of devices tested in vacuum or in air”.

The reason for the measurement at 0.25 Sun is because almost all the reported lateral single crystal devices are initially tested by vacuum probe station due to their fragility, and thus the highest light intensity is limited by equipment. The lateral structure PSC show potential attractive advantages; however, it was still on an initial stage with lower power conversion efficiency (PCE)

compared with regular PSC. Although we achieve nearly doubled PCE than previous report, we do not want to further claim the “record efficiency”.

To avoid misunderstanding and clarify device performance, we also show the light intensity dependent J - V curves for our lateral devices under AM1.5G illumination with different light intensities from 0.05 to 1.5 Sun by an AAA grade solar simulator, which is a comparable test condition to regular PSCs. And device still keep PCE of 11.52% under regular 1Sun condition for MAI doped device. Clearly the device performance is highly reliable and reproduceable and there was significant increase in device performance at different light intensity after MAI doping, which is not influenced much by test condition.

We add the light intensity (0.05 Sun to 1.5 Sun) dependent J - V curves and discussions in Page 14 Line 246 in the revised manuscript as: “Light intensity dependent J - V curves of lateral structured SC-PSCs under AM1.5G illumination with different light intensity from 0.05 to 1.5 Sun by an AAA grade solar simulator, which was a standard test condition for regular PSCs, was shown in Fig. 6A-B. The PSC based on as grown SCs showed much faster degradation rate in photocurrent density under higher light intensity (Fig. 5C). As a strong contrast, PSC with MAI doping kept liner relationship between photocurrent density and light intensity and a PCE of 11.52 % was achieved under 1 Sun condition and a champion PCE of 12.76 % under 0.5 Sun benefited from the suppressed surface recombination. In order to compared device performance with undoped SC-PSCs reported previously, which was always measured in the vacuum probe station for the fragile chip structured device with 0.25 sun illumination, J - V curves of devices (0.25 Sun) was shown in Fig. 6C.”

Reviewer #5 (Remarks to the Author):

25/08/2019

Review of the manuscript entitled "Efficient Lateral-structure Perovskite Single Crystal Solar Cells with High Operational Stability by Surface Doping" by Song et al., submitted to Nature Communications.

The manuscript reports on efficient and stable back-contact devices based on single crystal MAPbI₃ perovskites. Devices with efficiencies of up to 12.3% and operational stability over 1000 hrs are demonstrated. The authors show that by doping the perovskite film with MAI an enhanced energy level alignment with the anode is achieved, resulting in improved device efficiencies. The authors also attribute the improved device performance to a well passivated and highly conductive perovskite surface, which in turn leads to efficient charge collection.

This work is of definite interest to the perovskite community, specifically to those working on single crystals and back-contact structures. Throughout the manuscript, the authors present a thorough comparison between doped and un-doped devices. The main findings are clearly explained and the suggested methods for scalable device fabrication are remarkable.

Response: Thanks for the recognition of the significance of this work!

Comment 1: However, I find some of the discussion a bit difficult to read and understand. Therefore, I suggest the following revisions:

In the introduction (first paragraph), the authors mention wrong values of the efficiencies of the very first and latest reported perovskite solar cells. The correct values are 3.81% (not 3.9%, see Table 1, Kojima et al. 2009) and 25.2% (not 24.2%, please check the latest NREL chart, <https://www.nrel.gov/pv/assets/pdfs/best-research-cell-efficiencies.20190802.pdf>).

Response to C1: We have revised the paper as suggested.

Comment 2: In the introduction (second paragraph), the authors cite works on silicon solar cells employing the back-contact structure. Since reference 28 (Tainter et al.) is on perovskites, I think it should be removed from this sentence.

Response to C2: The reference has been removed here.

Comment 3: Another sentence must be added citing the following works on back-contact perovskite solar cells:

Response to C3: We have added a description in introduction part as: "Great efforts was made towards interdigitated back contact (IBC) structure perovskite solar cells (PSCs) due to its huge potential to realize high performance and low cost devices" and updated all of references below.

- Richard H. Friend, Felix Deschler, Luis M. Pazos-Outón, Mojtaba Abdi-Jalebi, Mejd Alsari, Back-Contact Perovskite Solar Cells. Scientific Video Protocols, 1, 1,

(2019), <https://doi.org/10.32386/scivpro.000005>, which provides an overview of back-contact perovskite solar cells;

- A. N. Jumabekov, E. Della Gaspera, Z. Q. Xu, A. S. R. Chesman, J. van Embden, S. A. Bonke, Q. Bao, D. Vak & U. Bach. Back-contacted hybrid organic-inorganic perovskite solar cells. *Journal of Materials Chemistry C* 4, 3125-3130, (2016). <https://doi.org/10.1039/C6TC00681G>

- Q. Hou, D. Bacal, A. N. Jumabekov, W. Li, Z. Wang, X. Lin, S. H. Ng, B. Tan, Q. Bao, A. S. R. Chesman, Y.-B. Cheng & U. Bach. Back-contact perovskite solar cells with honeycomb-like charge collecting electrodes. *Nano Energy* 50, 710-716, (2018). <https://doi.org/10.1016/j.nanoen.2018.06.006>

- Z. Hu, G. Kapil, H. Shimazaki, S. S. Pandey, T. Ma & S. Hayase. Transparent Conductive Oxide Layer and Hole Selective Layer Free Back-Contacted Hybrid Perovskite Solar Cell. *The Journal of Physical Chemistry C* 121, 4214-4219, (2017). <https://doi.org/10.1021/acs.jpcc.7b00760>

- A. N. Jumabekov, J. A. Lloyd, D. M. Bacal, U. Bach & A. S. R. Chesman. Fabrication of Back-Contact Electrodes Using Modified Natural Lithography. *ACS Applied Energy Materials* 1, 1077-1082, (2018). <https://doi.org/10.1021/acsaem.7b00213>

- Mejd Alsari, Oier Bikondo, James Bishop, Mojtaba Abdi-Jalebi, et al. In situ simultaneous photovoltaic and structural evolution of perovskite solar cells during film formation. *Energy & Environmental Science*, 11, 383, (2018), <https://doi.org/10.1039/C7EE03013D>

- Your ref. 28, Tainter et al. <https://doi.org/10.1016/j.joule.2019.03.010>

Comment 4: In the introduction (second paragraph), the sentence “The lateral structure was also a stand-free structure which did not require expensive indium tin oxide (ITO) substrate and did not have interfacial stability issues as that in traditional sandwiched-structure PSCs.” does not link very well with the previous sentence and therefore needs rewriting.

Response to C4: As suggested, we add a new discussion before this sentence as:

Page 3 Line 56: “Stand-free lateral structure single crystal perovskite solar cell (SC PSC) immune from strain stress induced mechanical device degradation due to different thermal expansion coefficients for multiple functional layers as well as substrates, which potentially increased the device stability than regular sandwich structures.”

Comment 5: In the results and discussions, the size of the back-contact devices is not clear from the discussion. I suggest the authors to show the sizes (electrodes spacing and electrodes sizes) in the schematics presented in Figure 1. I also suggest the authors to add another figure in Figure 1 where they show a schematic of the two-electrode devices to further clarify the difference between the small-area and large-area devices.

Response to C5: We have added the detailed definition of device area in Figure 1 and also in method part. Figure caption was revised as “(E) Structure of a regular single device with area of 50 μm x 1 mm. The large area device was integrated by 19 single devices (65 μm x 1.5 mm) with total device area of 1.85 mm^2 .”

Page 6 Line 110: Relative discussion was made as: “The width of each finger of electrode was approximately 50 μm and the spacing between each anode and cathode is approximately 65

μm in interdigital electrodes (Fig. 1A) and $50\ \mu\text{m}$ in a single device (Fig. 1E)” and “The single device area was $50\ \mu\text{m} \times 1\ \text{mm}$ and the 36 times scale up device area was $1.85\ \text{mm}^2$. A metal photomask was used during measurement to cover all the exposed crystal surface except working area and electrodes, while the working area was defined as the channel area between the metal electrodes.”

(E) Structure of a regular single device with area of $50\ \mu\text{m} \times 1\ \text{mm}$.

The large area device was integrated by 19 single devices ($65\ \mu\text{m} \times 1.5\ \text{mm}$) with total device area of $1.85\ \text{mm}^2$.

Comment 6: Table 1, an average of two values doesn't make statistical sense (columns SC-Cu and MAI/SC-Cu); please add the same amount of measurements used in the first two columns (SC-Au and MAI/SC-Au).

Response to C6: Thanks for the suggestion! As suggested, we add the additional data in revised Table 1.

Comment 7: Please revise XRD indexing in Figure 4, (check <https://pubs.rsc.org/en/content/articlelanding/2016/NJ/C6NJ00188B#!divAbstract>) and/or provide references.

Response to C7: We have revised XRD discussion part and cited this paper.

Comment 8: The authors mention ‘The SC-PSC devices showed nearly 50 times increase in rectifying ratio’. Wouldn't it be clearer if the authors mention just the FF? As an alternative can they comment on how this rectifying ratio is calculated?

Response to C8: Thanks for the comments, we have mention enhanced FF and add discussion in the revised paper as:

Page 12 Line 219: “There was significantly enhanced rectifying behavior in MAI treated SC-PSC device and brought a sharp increase in Fill Factors (FF) of PSCs from 41.6 % to 61.8 %”.

Comment 9: *The authors do not mention the active area of the solar cells as defined by the photomask.*

Response to C9: We have added the detailed definition of device area in Figure 1 and add a discussion as:

Page 21 Line 364: “A metal photomask was used during measurement to cover all the exposed crystal surface except working area and metal electrodes, while the working area is defined as the channel area between the metal electrodes. The PCE determination normalized by the active area for the interdigitated-structure devices.”

Comment 10: *In the results and discussions, the sentence “In ideal condition, the device performance would not be limited by the resistance of electrodes when scaled up, because almost all surface of the SCs were covered by metal electrode which would not significantly increase the series resistance of SC-PSCs compared with small area devices, which was an advantage of lateral-structure SC-PSCs compared with current sandwich structured PSCs and most of thin film solar cells.” is too long and hard to read and therefore needs rewriting.*

Response to C10: We have revised this discussion as:

Page 17 Line 299: “The lateral structure PSCs could achieve much lower series resistance than ITO or FTO based PSCs especially after scale up, because long range charge transport was realized by highly conductive metals for both anode and cathode in lateral structured SC devices, which potentially further promoted the fill factor and efficiency for PSCs”.

Comment 11: *Overall language/sentences throughout the manuscript need revising for better readability.*

Response to C11: Thanks for the suggestion, we have rechecked spelling and grammar in the paper and marked all changes in revised version.

Reviewers' comments:

Reviewer #2 (Remarks to the Author):

The authors have adequately revised their manuscript in response to my comments. I recommend the publication of this work.

Reviewer #3 (Remarks to the Author):

The authors addressed my previous questions and I am satisfied with the current version of the work.

Reviewer #4 (Remarks to the Author):

see attached document

Reviewer #5 (Remarks to the Author):

The authors have addressed all points in a satisfactory manner. The manuscript can be published as it is.

Reviewer #4 (Remarks to the Author):

The current manuscript shows best in class PCE with over 11% efficiency, much higher than previously reported efficiencies for any lateral contact device. The effect of MAI surface treatment is characterized and has an obvious and positive effect on the device performance.

I would recommend publishing the paper after some revisions to address the following issues.

1. The 1 sun efficiency, which is the figure of merit with regards to cell PCE, is 11.5% according to figure 6. Figures 6C-H still use the 0.25 sun data which I believe is inappropriate, and given the new 1 sun data, unnecessary. The 1 sun data is particularly critical when showing the short term stabilized power output (6D) and the long term stability (6H)
2. All claims to long term stability should be removed until the data is taken at 1 sun. There are many studies linking degradation activation to the electric field and light intensity. There is no reason to believe that the stability at 0.25 suns will hold under 1 sun illumination. This won't keep me from recommending publication of the paper, but it will keep me from recommending publication if the claims of "inspiring device stability" persist. This is a critical flaw that remains in the paper, claims of long term stability are simply not justified with the current data set.
3. On page 3 the authors claim that their devices "... did not have interfacial stability ..." issues as typical sandwich devices do, as with comment 2, this statement is not justified. Until the devices are tested for long term stability under 1 sun illumination the stability of the interfaces is not known. There are reports of voltage dependent electrochemistry even at noble metal contacts (10.1063/1.5083812), given the optical and electric fields in the device are lower at 0.25 suns than would be for 1 sun, the current data does not support the author's claims of long term stability of either the interfaces or the entire device.
4. The authors have added several pieces of data to show the impact of the MAI treatment on the crystal's measurable properties. However, they state the effect as being one of doping. I don't believe that the statement of doping is justified given their experimental results. I understand doping to be the addition of charge carriers to the lattice without a change in the band edges or band gap; i.e. doping changes the Fermi level but not the band edges. The data they present, particularly the KPFM, shows that the band edge location shifts and, according to the authors, gives a better alignment with the extracting contacts. I do believe that their data supports this claim as well as their claim to better surface passivation. However, better band alignment and better surface passivation are not the same as doping. I would like the authors to remove their language about doping and replace that with language that notes the better band alignment as these are two independent effects.
5. The authors refer to their structure as both lateral-structure and IBC (interdigitated back contact); references to IBC structure should be removed as their contacts were not on the back but on the illumination side of the device. This is important as there persists several challenges in moving from the architecture they used to a back contact device where illumination is not from the electrode side of the device. i.e. it's not a back contact device so they shouldn't call it that.

The authors have added some excellent work and data to the manuscript from its initial submission, in its current state the manuscript is of interest to the community and should be published after the revisions noted. I would also encourage the authors to look into the literature for lateral contact

devices as I believe 11.5% is the highest reported efficiency for any lateral contact device, not just single crystals.

Response Letter to reviewer:

The current manuscript shows best in class PCE with over 11% efficiency, much higher than previously reported efficiencies for any lateral contact device. The effect of MAI surface treatment is characterized and has an obvious and positive effect on the device performance.

I would recommend publishing the paper after some revisions to address the following issues.

Response: Thank you very much for the appreciate and the helpful suggestion to improve the quality of this work.

- 1. The 1 sun efficiency, which is the figure of merit with regards to cell PCE, is 11.5% according to figure 6. Figures 6C-H still use the 0.25 sun data which I believe is inappropriate, and given the new 1 sun data, unnecessary. The 1 sun data is particularly critical when showing the short term stabilized power output (6D) and the long term stability (6H)*

Response: Thanks for the comments, we have removed Fig. 6 C, D, H about the *J-V* curves and stability collected under 0.25 Sun and related discussion (Page 14). Instead of that, we provide 200 hours continues operational stability tracking data at **MPP condition under 1 Sun illumination**. There is no degradation observed during 200 hours long term operation, which is consistent with the result from that collected under 0.25 Sun. And *J-V* curves with forward and backward scan was added to show the little hysteresis effect with a discussion: “without obvious hysteresis effect observed (Fig. S1)” (Page 14, Line 252).

We revised description related to stability and updated the 1 Sun stability data to replace that under 0.25 Sun. Page 17 Line 304: “For long-term stability testing, the devices were continuously operated at MPP condition with 1 Sun illumination in glovebox without cooling stage and the photocurrent output were recorded. The device performance variation in 200 hours’ continuous operation was shown in Fig 6G. Inspiringly, our champion device still maintains ~100 % of its initial efficiency after operation at MPP 1 Sun condition for 200 hours.” In Abstract, and main test (Page 5 Line 89) we revised description to stability as: “Devices show excellent operational stability and no degradation observed after 200 hours’ continuous operation at maximum power point under 1 Sun illumination.”

- All claims to long term stability should be removed until the data is taken at 1 sun. There are many studies linking degradation activation to the electric field and light intensity. There is no reason to believe that the stability at 0.25 suns will hold under 1 sun illumination. This won't keep me from recommending publication of the paper, but it will keep me from recommending publication if the claims of “inspiring device stability” persist. This is a critical flaw that*

remains in the paper, claims of long term stability are simply not justified with the current data set.

Response: We have provided 200 hours continues operational stability at MPP condition under 1 Sun illumination and reorganized language to avoid misleading.

- 3. On page 3 the authors claim that their devices "... did not have interfacial stability..." issues as typical sandwich devices do, as with comment 2, this statement is not justified. Until the devices are tested for long term stability under 1 sun illumination the stability of the interfaces is not known. There are reports of voltage dependent electrochemistry even at noble metal contacts (10.1063/1.5083812), given the optical and electric fields in the device are lower at 0.25 suns than would be for 1 sun, the current data does not support the author's claims of long term stability of either the interfaces or the entire device.*

Response: We have removed the claim of "... did not have interfacial stability ..." in introduction because there is indeed no direct evidence in experiment was reported before this work can be published and revised it to: "The lateral structure was also a stand-free structure which did not require expensive indium tin oxide (ITO) substrate compared with traditional sandwiched-structure PSCs."

- 4. The authors have added several pieces of data to show the impact of the MAI treatment on the crystal's measurable properties. However, they state the effect as being one of doping. I don't believe that the statement of doping is justified given their experimental results. I understand doping to be the addition of charge carriers to the lattice without a change in the band edges or band gap; i.e. doping changes the Fermi level but not the band edges. The*

data they present, particularly the KPFM, shows that the band edge location shifts and, according to the authors, gives a better alignment with the extracting contacts. I do believe that their data supports this claim as well as their claim to better surface passivation. However, better band alignment and better surface passivation are not the same as doping. I would like the authors to remove their language about doping and replace that with language that notes the better band alignment as these are two independent effects.

Response: We agree with the comments about KPFM results, and we revised our claim about “due to the *p*-type self-doping effect of MAI” in revised manuscript (Page 6 Line 122).

We revise the “In addition to the doping effect” in Page 8 Line 146 to “In addition to the better band alignment”.

We use “MAI treatment” instead of “MAI doping” in revised manuscript and removed the “by surface doping” in the title.

We removed “doping effect and” at Page 12 Line 218.

We revised “Benefiting from the surface doping and passivation effect” to “Benefiting from the better band alignment and passivation effect” at Page 13 Line 243.

- 5. The authors refer to their structure as both lateral-structure and IBC (interdigitated back contact); references to IBC structure should be removed as their contacts were not on the back but on the illumination side of the device. This is important as there persists several challenges in moving from the architecture they used to a back contact device where illumination is not*

from the electrode side of the device. i.e. it's not a back contact device so they shouldn't call it that.

Response: To avoid mis-understanding to readers, we have revised the manuscript to clarify the device is lateral structure but not IBC structure yet. And we revised the claim of “Lateral structure was an emerging feasible structure for SC-PSCs and great efforts was made towards interdigitated back contact (IBC) structure PSCs due to its huge potential to realize high performance and low cost devices²⁶⁻³², which was well-studied in efficient silicon solar cells with outstanding PCE exceeding 26 %.^{33, 34}” to “Great efforts was made towards the emerging efficient lateral structure PSCs because it was believed as a prerequisite structure for efficient interdigitated back contact (IBC) structure PSCs to realize high performance and low cost devices.” and reference 33-34 was removed.

In conclusion section, we revised “With the development of large area thin single crystals growth and surface passivation technique, it will show a bright future and potentials for realizing IBC structured perovskite mono-crystalline solar cells, which could achieve efficiency comparable with that of the currently advanced IBC silicon solar cells at dramatically reduce material and fabrication cost.” To “With the development of large area thin single crystals growth and surface passivation technique, it will show a bright future and potentials towards efficient perovskite mono-crystalline solar cells with dramatically reduce material and fabrication cost.”

The authors have added some excellent work and data to the manuscript from its initial submission, in its current state the manuscript is of interest to the community and should be published after the revisions noted. I would also encourage the authors to look into the literature for lateral contact devices as I believe 11.5% is the highest reported efficiency for any lateral contact device, not just single crystals.

Response: Thanks again for the appreciate and the helpful suggestion to improve the quality of this work.

REVIEWERS' COMMENTS:

Reviewer #4 (Remarks to the Author):

The authors have made appropriate changes to address previous comments and can be published with no further modifications.